# LLM4EHR: Aligning Clinical Time Series with Medical Event Sequences via Large Language Models

## Abstract

Recent research in clinical machine learning, focusing on the intensive care unit (ICU), has shifted from bespoke supervised models to foundation models, utilising Large Language Models (LLMs). Here, LLMs are fine-tuned on mixtures of complex clinical data modalities, useful for various downstream tasks. However, existing methods do not sufficiently explore the shared temporal structure between the events on Electronic Health Records (EHRs) and clinical Time Series (TS) observations. This limitation potentially leads to less robust and adaptive clinical foundation models, resulting in reduced performance on downstream tasks. To fully exploit this temporal structure, we propose LLM4EHR, a new clinical foundation model trained on ICU data. Combining pre-trained LLMs with additional trainable layers, we fine-tune our model to temporally align the EHR and TS modalities. For this, we propose a regularised contrastive objective to jointly learn representations of EHRs and clinical TS. Supported by an ablation study, we find that embeddings from LLM4EHR improve performance on various downstream clinical tasks with competitive performance in a few-shot setting. Further, we empirically demonstrate that LLM4EHR learns transferable clinical TS embeddings that can be deployed to new cohorts with minimal performance loss. These findings provide a step towards building more generalisable and performant clinical foundation models.

## 1 Introduction

Electronic Health Records (EHRs) data is inherently multimodal. Temporal patient observations, interactions and measurements are recorded as structured EHR events and numerical clinical Time Series (TS), as shown in **Figure 1 a**. Specifically, EHR events consist of coded clinical interactions aligned with timestamps, while patient clinical TS include numerical measurements for monitoring data such as vital signs and physiological signals. These two data modalities complement each other in representing a patient's health. However, the structural differences between these modalities pose a significant challenge for building versatile clinical foundation models.

Towards overcoming this, we release a new clinical foundation model, LLM4EHR, which jointly models EHR event sequences and clinical TS observations whilst respecting their temporal relationships. We find that the resulting learnt representations can adapt to downstream clinical tasks such as mortality prediction, phenotyping and remaining Length of Stay (LoS) prediction, which we support with an ablation study. Additionally, we show that LLM4EHR can be adapted to new datasets using shared clinical TS variables with consistent performance.

Traditionally, bespoke supervised methods were developed individually for clinical tasks, including survival analysis (Ghassemi et al., 2014; Barajas & Akella, 2015), risk prediction (Alvarez et al., 2013; Cheng et al., 2016) and phenotyping (Albers et al., 2014; Liu et al., 2015). However, experiments conducted on large multitask clinical benchmarks (Harutyunyan et al., 2019; Sheikhalishahi et al., 2020) demonstrated that learnt information from one task is often relevant for others. Furthermore, patient observations in the clinical setting form temporal sequences, making them well-suited for language modelling techniques. These observations, along with recent findings that Large Language Models (LLMs) function as unsupervised multitask learners (Radford et al., 2019), have driven efforts to develop multitask clinical foundation models (Wornow et al., 2023). Recent work has thus explored

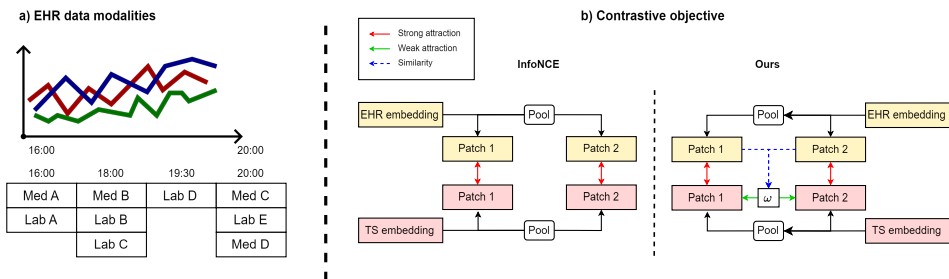

Figure 1: a) An example of the two temporal EHR data modalities and b) the difference between the InfoNCE (van den Oord et al., 2018; He et al., 2020) and our regularised contrastive objective, both the TS and EHR embeddings are processed by the pre-trained LLM attentions before being temporally pooled into patches for calculating calculating the contrastive objective, we used average pooling to create vector representations of TS and EHR patches.

using pre-trained LLMs for modelling EHR event trajectories (Li et al., 2020; Rasmy et al., 2021; Renc et al., 2024; Kraljevic et al., 2024) or clinical TS (Chang et al., 2025; Liu et al., 2024). A review by Wornow et al. (2023) found that clinical foundation models, pre-trained on large unlabelled EHR datasets, can adapt to downstream tasks using limited labelled data.

However, many current methods fail to incorporate multiple clinical data modalities during pre-training, despite evidence that combining modalities such as clinical notes and TS improves down-stream performance (King et al., 2023; Chan et al., 2024; An et al., 2021). Given the interdependency between EHR events and clinical TS observations, we hypothesise that similar improvements could be achieved by jointly modelling these two temporal modalities via self-supervised pre-training.

Prior work aligning these different clinical data modalities focused on contrastive representation learning between pairs of samples (Kline et al., 2022). In these cases, the alignment is often centred on shared identities between modalities, such as aligning the clinical TS and note embeddings collected in the same care period (King et al., 2023; An et al., 2021). However, this stationary, instance-wise alignment across clinical data modalities does not consider the shared temporal structure between EHR events and clinical TS observations, which we hypothesise provide more effective learnt representations.

In this work, we introduce LLM4EHR, a multimodal clinical foundation model that jointly models EHR sequences and clinical TS observations. Unlike prior work, LLM4EHR learns entangled clinical TS representations with EHR sequence embeddings produced by a pre-trained LLM. We reformulate the commonly used contrastive objective to temporally align the TS observations with EHR event sequences in the shared latent space. This leads to improved performance in downstream tasks that generalise to new datasets.

Additionally, we propose utilising pre-trained LLM embeddings to mitigate the effects of class collision (Chen et al., 2020; Zheng et al., 2021; Wu et al., 2024) in contrastive learning. We present a regularised InfoNCE loss (van den Oord et al., 2018; He et al., 2020), where the semantic similarities between LLM-embedded EHR events are used to weight the learnt clinical TS embeddings (**Figure 1 b**), providing improved embeddings. Our contributions are as follows:

- We present LLM4EHR, an LLM-based framework for representation learning of clinical data, trained on data from intensive care units.

- To train LLM4EHR, we propose a contrastive learning objective to temporally align TS and EHR sequences. With an ablation study, we find that this improves the learnt representations.

- We show that our method achieves superior performance in clinical classification tasks in few-shot and cross-dataset settings, including in the presence of data distribution shift.

## 2 RELATED WORK

EHR foundation models are clinical foundation models trained on large-scale EHR data (Wornow et al., 2023). Recent methods exhibit notable similarities with LLMs in that they are trained on tokenised event sequences via autoregressive (Renc et al., 2024; Kraljevic et al., 2024; 2021; Cascella et al., 2023) or masked sequence modelling (Rasmy et al., 2021; Li et al., 2020; Alsentzer et al., 2019; Pang et al., 2021). Notable examples include Foresight by Kraljevic et al. (2024) and ETHOS by Renc et al. (2024), which repurposed the autoregressive attention layers of GPT-2 (Radford et al., 2019) to simulate event transitions in patient timelines. However, these works learn representations of EHRs separately from TS observations, which can provide important supplementary information.

Self-supervised representation learning for TS learns transferable representations of TS features from unlabelled data for downstream tasks (Dong et al., 2023; Nie et al., 2023; Ekambaram et al., 2023). Recent work demonstrated improved performance in TS forecasting by modelling TS via pre-trained LLM token transitions. Here, TS segments are tokenised to form temporal sequences, and LLMs are repurposed to process TS tokens via self-supervised pretext tasks. LLM4TS by Chang et al. (2025) used self-supervised reconstruction to repurpose pre-trained LLMs for TS forecasting. Similarly, AutoTimes by Liu et al. (2024) used language prompts during self-supervised pre-training to allow in-context TS forecasting via frozen LLMs. However, prior works do not consider the temporal relationship between EHRs and TS observations, which can be jointly represented.

Contrastive learning can create entangled representations between pairs of samples. Prior works in clinical machine learning explored aligning similar clinical TS samples via contrastive learning. EBCL by Jeong et al. (2024) used contrastive pre-training to align TS observations before and after key index events and showed improved performances in downstream classification tasks. CROCS by Kiyasseh et al. (2021) used contrastive learning to align ECG measurement embeddings with patient prototype embeddings, where patient prototypes serve as queries to retrieve similar ECG samples. King et al. (2023) explored aligning clinical time series (TS) with ICU notes via a shared LLM. However, their work maintains a stationary alignment between TS and clinical notes and does not consider important temporal relationships.

## 3 BACKGROUND

### 3.1 CLINICAL TIME SERIES AND EHR RECORDS

When patients interact with care services, each measurement, observation, and test is recorded in an Electronic Health Record (EHR) or as part of a clinical Time Series (TS). Clinical time series (TS) consist of regularly collected patient physiological measurements and vital signs, and are collected during a defined period of stay. Clinical TS observations are used to monitor a patient's health status and contribute to standardised risk analysis such as SAPS (Le Gall et al., 1984) or SOFA (Jones et al., 2009). Supervised methods in clinical machine learning use TS features as input for downstream tasks, such as survival analysis (Ghassemi et al., 2014; Barajas & Akella, 2015), risk predictions (Alvarez et al., 2013; Cheng et al., 2016) and phenotyping (Albers et al., 2014; Liu et al., 2015). In contrast, EHR events are itemised clinical observations and interactions aligned with time. EHR events are coded following a hospital's internal coding system (Johnson et al., 2016; Pollard et al., 2018). Typically, EHR events are tokenised to form temporal sequences for sequence modelling (Steinberg et al., 2024). Here, predicting future events are used to simulate a patient's future health trajectory (Renc et al., 2024; Kraljevic et al., 2024).

Formally, clinical time series form multivariate numerical time series consisting of regularly collected patient observations. For this work, we define an instance of clinical TS with $L$ variables over $T$ time steps as $x^{1:T} = \{x^1, x^2, \cdots, x^T\} \in \mathbb{R}^{T \times L}$.

An EHR sequence, however, will be represented as a stream of tokens. This is defined as $k^{1:T} = \{k_1^1, k_2^1, \cdots, k_N^T\}$, where $k_j^t \in \mathcal{K}$ represents $j^{th}$ tokenised clinical events recorded at time step $t$. This token set $\mathcal{K}$ refers to natural language phrases, such as drug names or lab test components. New token embeddings, not previously found in the LLM's tokeniser, are initialised by averaging the weights of its text descriptors.

In our work, we consider data collected for a given patient throughout a single Intensive Care Unit (ICU) stay as an 'episode'. A full episode is defined as $e_i = [x_i, k_i]$, where both $x_i$ and $k_i$ are centred on ICU admission and are aligned with the time indicator $t \in T$.

## 3.2 REPRESENTATION LEARNING FOR CLINICAL TIME SERIES

Both EHRs and TS provide considerable context of a patient's health that could allow for the design of predictive models for predicting clinical events. Before recently, bespoke predictive models were built for each predictive task, often leading to duplicated efforts and sub-optimal predictive performance. Recent advances in LLMs and methods for contrastive learning have brought the onset of clinical foundation models, which provide a unified method to encode clinical data.

However, current methods encode EHRs and TS without considering the alignment of the time at which events are recorded in each sequence. This motivates the exploration of methods that align both sequences in time as part of the representation learning process.

## 4 METHODS

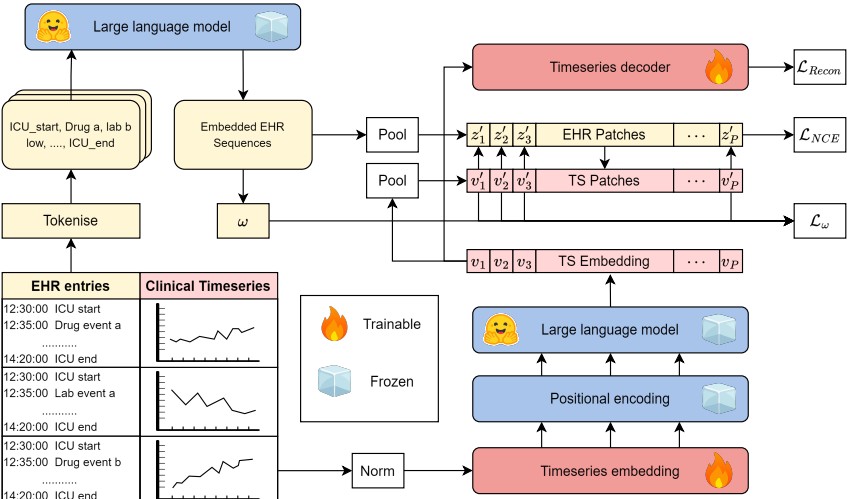

Figure 2: Overview of LLM4EHR, EHR entries and clinical TS are centred on ICU admissions, where EHR operations are coloured in beige and TS operations are coloured in red, pre-trained LLM layers (besides the embedding weights for new tokens) are frozen during pre-training.

## 4.1 LLM4EHR: LEARNING CROSS-MODALITY ALIGNMENT OF EHR DATA

By temporally aligning learnt representations of EHR and TS data, existing in different modalities, we aim to improve the use of machine learning on clinical tasks. For this, we propose LLM4EHR, a pre-training framework designed to adapt LLMs for modelling temporal clinical data (**Figure 2**).

EHR sequences $k_i$ and clinical TS $x_i$ complement each other in representing a patient's ICU stay $e_i$. Consequently, incorporating information from EHR sequences leads to more generalisable TS embeddings for downstream clinical tasks, such as classification and regression. We formulate a contrastive objective to align the two sequences in the time domain, which pulls the embeddings of TS observations and EHR token embeddings closer at time step $t$ for each episode.

Aligning the TS and EHR sequence embeddings ($v_i$ and $z_i$, respectively) step-by-step is problematic due to the structural differences between EHR sequences and clinical TS. EHR data are usually collected at a lower frequency than TS observations and measurements. For example, while vital signs such as heart rate are collected hourly by care providers, medication events are only recorded when the rate of intake changes. Pulling the $v^t$ uniformly towards all token embeddings at step $t$ creates an ambiguous learning objective, where $v^t$ is pulled towards multiple or zero instances of $z_i^t$.

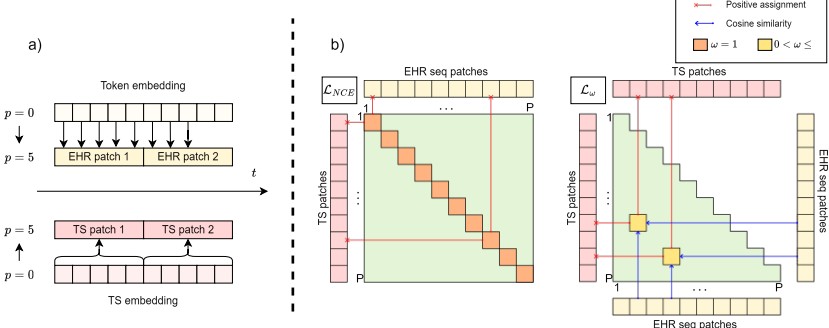

Figure 3: a) patch aggregation and b) calculating the semantic alignment loss; the figures only include the lower triangular and non-diagonal part of the soft assignment $\omega$ since $\omega$ is symmetric.

Additionally, uniformly pulling TS embedding towards individual token embeddings could skew the agreement between TS and EHR sequence embeddings towards frequently occurring tokens.

PATCH AGGREGATION

We propose to temporally aggregate $v_i$ and $z_i$ into patches before calculating the contrastive objective. For this, we define patches as temporal bins, where the TS and EHR sequence embeddings are aggregated into equal numbers of non-overlapping patches. Embeddings within patches are pooled to create vector representations for the TS and EHR sequence patches. In this case, the contrastive objective is explicitly defined between pairs of TS and EHR sequence patches. These operations are shown in **Equation 1** and **Figure 3 a**, where $p$ and $s$ are the filter size and stride in the temporal dimension. Here, we set $p = s$ to create non-overlapping patches.

$$v_i^{'} = \mathrm{AvgPool}(v_i \mid p, s) \qquad\qquad z_i^{'} = \mathrm{AvgPool}(z_i \mid p, s) \qquad (1)$$

SEMANTIC ALIGNMENT

The attraction between two instances of $v_i'$ and $z_i'$ is defined patch-wise via a modified InfoNCE (van den Oord et al., 2018) loss. Similar to He et al. (2020) and Kiyasseh et al. (2021), we implemented the normalised variant of InfoNCE $\mathcal{L}_{\mathrm{NCE}}$ (van den Oord et al., 2018), with temperature $\tau$ (defined in Appendix A.1). However, we note that this objective does not address the issue of class collision, where instances within same class are pushed apart due to the loss being defined between an instance and its augmented view. In our case, similar instances of TS observations should remain closer after embedding. Since LLMs are known to capture word similarities (Mahajan et al., 2025), we propose a $\omega$-regularised variant of the InfoNCE loss to weight $v_i^{'}$ based on the semantic similarities between instances of $z_i^{'}$. Defining sim() as the cosine similarity, the regularisation function $\mathcal{L}_{\omega}$ is given as:

$$\mathcal{L}_{\omega} = -\sum_{i=1}^{P}\sum_{\substack{j=1 \\ j<i}}^{P} \omega_{i,j} \log \frac{\exp(\mathrm{sim}(v_i', v_j')/\tau)}{\sum_{k\in P\setminus i}\exp\left(\mathrm{sim}(v_i', v_k')/\tau\right)} \quad \omega_{i,j} = \frac{\exp(\mathrm{sim}(z_i', z_j'))}{\sum_{k\in P\setminus i}\exp(\mathrm{sim}(z_i', z_k'))} \quad (2)$$

The attraction weight $\omega_{i,j}$ is interpreted as the degree of agreement between $v_i'$ and $v_j'$, quantified using the similarity between $z_i'$ and $z_j'$. For a pair of $v_i'$ and $z_i'$, the weight $\omega$ is a $P \times P$ symmetric matrix, as shown in **Figure 3 b**. We thus used only the lower triangular part of $\omega$, such that $\omega_{i,j}$ is only defined for $i < j$. Intuitively, **Equation 2** pulls TS observations corresponding to similar EHR events closer, such that similar instances within $x_i$ are consistent after embedding. The overall alignment objective is defined as:

$$\mathcal{L}_{\mathrm{align}} = \mathcal{L}_{\mathrm{NCE}} + \mathcal{L}_{\omega} \qquad (3)$$

THE ROLE OF LLM

As shown in **Figure 2**, we used the LLM's frozen attention layers as a consistent mechanism for modelling the transitions of EHR events and TS observations. The LLM's attention and embedding layers are kept frozen to ensure that $z_i$ remains consistent for optimising $\mathcal{L}_{\text{align}}$ (He et al., 2020). Similar to Chang et al. (2025) and Liu et al. (2024), we treat $x \in x^{1:T}$ as tokens and employ pre-trained autoregressive attention mechanisms from GPT-2 (Radford et al., 2019) to model the transition between TS observations. Inspired by Lee et al. (2024), by aggregating $v_i'$ after the attention modelling, LLM4EHR can make dynamic downstream predictions, such as an hourly mortality forecast. To produce our final optimisation objective, we reconstruct the input TS autoregressively from the latent representation $v_i$ and use the reconstruction loss as an auxiliary objective to $\mathcal{L}_{\text{align}}$. This provides the loss value:

$$\mathcal{L}_{\text{total}} = \mathcal{L}_{\text{align}} + \mathcal{L}_{\text{recon}} = \mathcal{L}_{\text{align}} + \frac{1}{T}\sum_{t=2}^{T}\|x_i^t - \hat{x_i}^t\|_2^2 \tag{4}$$

## 5 RESULTS

### 5.1 EXPERIMENTAL SETTING

We employ four downstream clinical prediction tasks to evaluate the performance of our pre-trained EHR foundation model. We pre-trained our model using two public intensive care EHR datasets: MIMIC-III (Johnson et al., 2016) and eICU (Pollard et al., 2018). We then evaluated our model using pre-defined labels for 48 hours in-unit mortality, phenotyping, decompensation and remaining Length of Stay (LoS) predictions. The labelled TS episodes were extracted following the pipelines created by Harutyunyan et al. (2019) and Sheikhalishahi et al. (2020) for MIMIC-III and eICU, respectively. While the original benchmarks were fully labelled, we re-partitioned the datasets to be 70% for self-supervised pre-training, 20% for fine-tuning and 10% for testing. The re-partitioning was designed to simulate the deployment of LLM4EHR, where the model is trained on a large unlabelled dataset and adapted to downstream tasks using limited labelled data. We evaluate the transferability of our pre-trained model on two task-specific EHR datasets: the Physionet Challenge 2012 (Physio2012) (Silva et al., 2012) and a private hospital paediatric ICU dataset (PICU). **Table 1** provides a summary of the datasets.

Table 1: Overview of the number of samples in each evaluation dataset and their use. Here, ✓ indicates the availability of task labels for the dataset.

| Datasets | MIMIC-III | eICU | Physio2012 | PICU |
|---|---|---|---|---|
| Pre-training | 28588 | 21000 | N/A | N/A |
| Fine-tuning | 8168 | 6000 | 4000 | 1050 |
| Mortality | ✓ | ✓ | ✓ | ✓ |
| Phenotyping | ✓ | ✓ | ✗ | ✗ |
| Decompensation | ✓ | ✓ | ✗ | ✗ |
| Remaining LoS | ✓ | ✓ | ✗ | ✗ |
| Testing | 4085 | 3000 | 4000 | 250 |

We evaluated the performance of LLM4EHR against other TS models in a few-shot scenario, since labelled instances are rare in real-world settings (Xiao et al., 2018). The baseline models are grouped into three categories: **(1) Generic models**, including three common supervised deep learning structures for TS modelling: Multilayer Perceptrons (MLP), Bidirectional Long Short-Term Memory Model (LSTM) and Convolutional Neural Networks (CNN). **(2) TS models**, including three self-supervised self-attention models for general TS modelling: SimMTM (Dong et al., 2023), PatchTST (Nie et al., 2023) and PatchTSMixer (Ekambaram et al., 2023). Inspired by prior research such as Liu et al. (2024) and Chang et al. (2025), we also included an ablated version of our pre-trained model, the supervised GPT (Radford et al., 2019). **(3) EHR models**, including four transformer-based models specifically developed for modelling EHR data: EBCL (Jeong et al., 2024) aligns TS embedding before and after key index events for given patients; King et al. (2023) presented a transformer-based contrastive pre-training pipeline to align clinical notes and TS embedding for individual patients; similarly, CTPD (Wang et al., 2025) is a transformer-based pipeline for learning cross-modal temporal

patterns between clinical notes and TS embedding; and SAnD (Song et al., 2018) is a supervised autoregressive attention model for clinical TS that generalises to various downstream prediction tasks. **Table 2** summarises key qualities of these baseline methods.

Table 2: Overview of the baseline models, unless otherwise specified, LLM4EHR uses pre-trained GPT-2 (Radford et al., 2019) as the backbone.

|  |  | Generic Models | Supervised GPT | SimMTM | PatchTSMixer | PatchTST | EBCL | King et al. (2023) | CTPD | SAnD | LLM4EHR |
|---|---|---|---|---|---|---|---|---|---|---|---|
| Pre-training | MIMIC-III | ✗ | ✗ | ✓ | ✓ | ✓ | ✓ | ✓ | ✗ | ✗ | ✓ |
|  | eICU | ✗ | ✗ | ✓ | ✓ | ✓ | ✗ | ✗ | ✗ | ✗ | ✓ |
| Backbone | Self-attention | ✗ | ✓ | ✓ | ✓ | ✓ | ✓ | ✓ | ✓ | ✓ | ✓ |
|  | LLM | ✗ | ✓ | ✗ | ✗ | ✗ | ✗ | ✓ | ✓ | ✗ | ✓ |
| Modality | TS | ✓ | ✓ | ✓ | ✓ | ✓ | ✗ | ✓ | ✓ | ✓ | ✓ |
|  | EHR-Sequence | ✗ | ✗ | ✗ | ✗ | ✗ | ✓ | ✗ | ✓ | ✗ | ✓ |
|  | Clinical notes | ✗ | ✗ | ✗ | ✗ | ✗ | ✗ | ✓ | ✓ | ✗ | ✗ |

## 5.2 FEW-SHOT EVALUATION ON MIMIC-III

The clinical prediction benchmarks Harutyunyan et al. (2019) and Sheikhalishahi et al. (2020) defined four clinical prediction tasks using routinely collected TS data from ICU admissions: (1) Mortality prediction uses the first 48 hours of patient measurements to predict the risk of in-unit mortality; (2) Phenotyping classifies patients against pre-defined acute phenotypes; (3) Decompensation predicts immediate declines in patients' physiological states, characterised by their risk of in-unit mortality in the next 24 hours; and (4) Remaining LoS is a regression task to predict the hours until ICU discharge for a patient. Decompensation and remaining LoS predictions were made hourly, and we evaluated the remaining LoS predictions in days, as in Sheikhalishahi et al. (2020). Differing from this work, we truncated the patient TS and EHR sequences at 200 hours. Additionally, due to overlapping labels between mortality and decompensation predictions, we reserved the 48-hour mortality prediction for cross-dataset evaluations.

Table 3: Evaluation metrics for all fine-tuning tasks in MIMIC-III (Johnson et al., 2016), the results were averaged over ten runs and presented as mean (std), best results are indicated in bold.

| MIMIC-III | | | | | | |
|---|---|---|---|---|---|---|
| Tasks | | Phenotyping | | Decompensation (hourly) | | Remaining LoS | |
| Models | | AUC-ROC (macro) | AUC-ROC (micro) | AUC-ROC | AUC-PRC | RMSE | R2 |
| Generic Models | MLP | 0.622 (0.022) | 0.728 (0.031) | 0.816 (0.090) | 0.105 (0.052) | 6.872 (0.028) | -0.237 (0.010) |
|  | CNN | 0.680 (0.005) | 0.742 (0.005) | N/A | N/A | N/A | N/A |
|  | LSTM | 0.661 (0.005) | 0.739 (0.004) | 0.899 (0.033) | 0.309 (0.017) | 5.919 (0.062) | 0.082 (0.019) |
| TS Models | Supervised GPT | 0.654 (0.004) | 0.717 (0.021) | 0.877 (0.009) | 0.235 (0.018) | 6.012 (0.088) | 0.046 (0.095) |
|  | SimMTM | 0.708 (0.004) | 0.757 (0.033) | 0.899 (0.015) | 0.325 (0.041) | **5.695 (0.072)** | **0.156 (0.045)** |
|  | PatchTSMixer | 0.656 (0.003) | 0.737 (0.003) | 0.901 (0.008) | 0.328 (0.022) | 5.824 (0.076) | 0.053 (0.025) |
|  | PatchTST | 0.651 (0.002) | 0.737 (0.003) | 0.894 (0.015) | 0.280 (0.032) | 5.722 (0.085) | 0.135 (0.027) |
| EHR Models | EBCL | 0.694 (0.003) | 0.743 (0.014) | N/A | N/A | N/A | N/A |
|  | King et al. | 0.702 (0.004) | 0.756 (0.005) | N/A | N/A | N/A | N/A |
|  | CTPD | 0.707 (0.003) | 0.755 (0.004) | N/A | N/A | N/A | N/A |
|  | SAnD | 0.655 (0.011) | 0.721 (0.014) | 0.872 (0.031) | 0.298 (0.045) | 5.873 (0.066) | 0.076 (0.077) |
| Ours | LLM4EHR | **0.719 (0.003)** | **0.761 (0.006)** | **0.912 (0.003)** | **0.333 (0.025)** | 5.743 (0.088) | 0.129 (0.054) |

The results in **Table 3** show that our LLM4EHR consistently outperforms baseline models in classification tasks while achieving competitive performance in predicting the remaining LoS. Similar trend can be observed in few-shot experiments with 10% and 5% fine-tuning data, as shown in **Section A.10**. Our model underperformed against recent TS models for predicting LoS, suggesting that our multimodal pre-training framework is better suited for downstream classification tasks. We hypothesise that this indicates distortions in LLM4EHR's learned TS feature embeddings, which we further investigate in **Section A.5**.

## 5.3 CROSS-DATASET MORTALITY PREDICTION

Next, we evaluated the transferability of our pre-trained model via cross-dataset mortality predictions. As mortality labels are universally available across the four ICU datasets, we used models developed from MIMIC-III (Johnson et al., 2016) and eICU (Pollard et al., 2018) to predict ICU mortalities for new patient cohorts. Here, the numerical features in the target dataset were normalised based on their mean and variance in the source dataset. Since PICU contains only paediatric patients, we

note a significant data distribution shift from MIMIC-III. Despite this, the results in **Table 4** illustrate that our pre-trained model adapts well to new patient cohorts, even in cases with significant data distribution shifts.

Table 4: Results for cross-dataset mortality predictions, where all models except the generic models were pre-trained on the source dataset before being deployed to the target dataset. We reported the challenge scores for the Physionet2012 (Silva et al., 2012) dataset, and the best results are indicated in bold.

| Datasets | | MIMIC-III → Physionet 2012 | | MIMIC-III → PICU | | eICU → MIMIC-III | | MIMIC-III → eICU | |
|---|---|---|---|---|---|---|---|---|---|
| Models | | Sensitivity score (Best/Mean) | H score (Best/Mean) | AUC-ROC (Mean/std) | AUC-PRC (Mean/std) | AUC-ROC (Mean/std) | AUC-PRC (Mean/std) | AUROC (Mean/std) | AUPRC (Mean/std) |
| Generic Models | MLP | 0.425 (0.399) | 66.046 (88.432) | 0.738 (0.007) | 0.212 (0.024) | 0.801 (0.002) | 0.395 (0.014) | 0.841 (0.008) | 0.393 (0.012) |
| | CNN | 0.440 (0.416) | 67.445 (122.907) | 0.787 (0.044) | 0.183 (0.031) | 0.833 (0.017) | 0.534 (0.032) | 0.863 (0.007) | 0.451 (0.018) |
| | LSTM | 0.468 (0.457) | 31.194 (49.900) | 0.822 (0.019) | **0.317 (0.050)** | 0.838 (0.005) | 0.581 (0.015) | 0.888 (0.009) | 0.363 (0.013) |
| TS Models | Supervised GPT | 0.424 (0.369) | 48.578 (60.982) | 0.704 (0.022) | 0.091 (0.007) | 0.832 (0.003) | 0.529 (0.009) | 0.893 (0.007) | 0.435 (0.007) |
| | SimMTM | 0.459 (0.447) | 33.802 (54.355) | 0.843 (0.009) | 0.215 (0.015) | 0.869 (0.004) | 0.594 (0.013) | 0.891 (0.014) | 0.447 (0.023) |
| | PatchTSMixer | 0.432 (0.429) | 42.073 (37.524) | 0.789 (0.045) | 0.202 (0.031) | 0.800 (0.009) | 0.457 (0.018) | 0.859 (0.016) | 0.390 (0.017) |
| | PatchTST | 0.375 (0.364) | 70.577 (144.676) | 0.748 (0.070) | 0.242 (0.042) | 0.764 (0.015) | 0.400 (0.034) | 0.820 (0.024) | 0.319 (0.032) |
| EHR Models | EBCL | 0.399 (0.292) | 82.433 (152.712) | 0.779 (0.019) | 0.199 (0.013) | 0.829 (0.017) | 0.514 (0.032) | 0.897 (0.003) | 0.434 (0.009) |
| | King et al. | 0.438 (0.400) | 59.190 (102.434) | 0.794 (0.021) | 0.183 (0.033) | N/A | N/A | 0.884 (0.005) | 0.372 (0.007) |
| | CTPD | 0.457 (0.414) | 39.221 (50.042) | 0.824 (0.017) | 0.227 (0.021) | N/A | N/A | 0.886 (0.004) | 0.385 (0.008) |
| | SAnD | 0.414 (0.293) | 109.423 (155.981) | 0.802 (0.019) | 0.205 (0.028) | 0.842 (0.007) | 0.524 (0.021) | 0.891 (0.011) | 0.427 (0.009) |
| Ours | LLM4EHR | **0.483 (0.474)** | **25.814 (38.733)** | **0.849 (0.008)** | 0.274 (0.016) | **0.877 (0.006)** | **0.595 (0.022)** | **0.917 (0.015)** | **0.512 (0.025)** |

## 5.4 FURTHER ANALYSIS OF LLM4EHR

### ABLATION STUDY OF THE PRE-TRAINING OBJECTIVE

We experimented with different pre-training objectives to assess the impact of our $\omega$-regularised contrastive objective. As shown in **Table 5**, $\mathcal{L}_\omega$ improves the performance of predicting decompensation by 26%, along with marginal improvements in phenotyping and predicting remaining LoS. The significant improvement in predicting decompensation demonstrates the regularising effect of $\mathcal{L}_\omega$ in learning consistent TS embeddings for dynamic classification. Additionally, models pre-trained without the reconstruction objective are significantly worse at predicting remaining LoS. This shows that the reconstruction objective is crucial for preserving numerical features in the learned TS embeddings. **Table 5** also shows that masking $\mathcal{L}_\omega$ had minimal effects on the model's performance. However, using the masked $\omega$ is preferable as $\omega$ is symmetric.

Table 5: The effects of different pre-training objectives on the downstream task, where 'full' means the full $\omega$ was used as opposed to the masked version in **Section 4**, best results are highlighted in bold.

| Tasks | Phenotyping | | Decompensation (hourly) | | Remaining LoS | |
|---|---|---|---|---|---|---|
| Loss | AU-ROC (marco) | AU-ROC (micro) | AU-ROC | AU-PRC | RMSE | R2 |
| $\mathcal{L}_{\text{NCE}}$ | 0.686 (0.001) | 0.749 (0.004) | 0.867 (0.007) | 0.245 (0.021) | 6.067 (0.056) | 0.028 (0.152) |
| $\mathcal{L}_{\text{NCE}} + \mathcal{L}_{\text{recon}}$ | 0.701 (0.001) | 0.756 (0.002) | 0.873 (0.005) | 0.257 (0.015) | 5.865 (0.064) | 0.126 (0.096) |
| $\mathcal{L}_{\text{NCE}} + \mathcal{L}_\omega$ | 0.696 (0.002) | 0.744 (0.003) | 0.887 (0.006) | 0.275 (0.017) | 6.011 (0.073) | 0.045 (0.052) |
| $\mathcal{L}_{\text{NCE}} + \mathcal{L}_{\omega(\text{full})} + \mathcal{L}_{\text{recon}}$ | 0.717 (0.004) | 0.758 (0.004) | 0.911 (0.007) | 0.313 (0.015) | **5.740 (0.072)** | **0.135 (0.044)** |
| $\mathcal{L}_{\text{NCE}} + \mathcal{L}_\omega + \mathcal{L}_{\text{recon}}$ | **0.719 (0.003)** | **0.761 (0.006)** | **0.912 (0.003)** | **0.333 (0.025)** | 5.743 (0.088) | 0.129 (0.054) |
| $\mathcal{L}_{\text{recon}}$ | 0.672 (0.002) | 0.748 (0.002) | 0.861 (0.006) | 0.223 (0.010) | 5.792 (0.041) | 0.133 (0.031) |

### THE EFFECTS OF THE TEMPERATURE $\tau$

The temperature $\tau$ for the contrastive objective and the patch size for aggregating the TS and EHR patch embeddings are two crucial hyperparameters in LLM4EHR. We note that the choice of patch size is limited in practice due to the relative sparsity of EHR data compared to TS observations. We aggregated the TS and EHR embeddings in a five-hour window to ensure most patches in a batch of 64 samples would contain relevant EHR entries for calculating the contrastive loss. For $\tau$, we evaluated its effects on the pre-trained models by measuring their performance in the downstream tasks. **Table 6** shows that a smaller $\tau$ is overall better for pre-training, with a noticeable drop in performance between $\tau = 0.20$ and $\tau = 0.50$.

Table 6: The effects of the contrastive temperature $\tau$ on the downstream task, since the optimal $\tau$ was different for each task, we selected $\tau = 0.02$ for all evaluations as it achieved competitive results in all three tasks.

| Tasks | Phenotyping | | Decompensation | | Remaining LoS | |
|---|---|---|---|---|---|---|
| $\tau$ | AU-ROC (macro) | AU-ROC (micro) | AU-ROC | AU-PRC | RMSE | R2 |
| $\tau = 0.02$ | **0.719 (0.003)** | **0.761 (0.006)** | 0.912 (0.003) | 0.333 (0.025) | 5.743 (0.088) | 0.129 (0.054) |
| $\tau = 0.08$ | 0.707 (0.005) | 0.755 (0.007) | **0.914 (0.003)** | **0.335 (0.021)** | 5.790 (0.112) | 0.115 (0.071) |
| $\tau = 0.14$ | 0.692 (0.008) | 0.744 (0.012) | 0.907 (0.002) | 0.335 (0.014) | **5.720 (0.079)** | **0.136 (0.060)** |
| $\tau = 0.20$ | 0.701 (0.007) | 0.748 (0.006) | 0.895 (0.005) | 0.274 (0.014) | 5.875 (0.128) | 0.089 (0.084) |
| $\tau = 0.50$ | 0.685 (0.008) | 0.750 (0.007) | 0.876 (0.009) | 0.238 (0.013) | 5.802 (0.133) | 0.111 (0.088) |

THE CHOICE OF LLMs FOR EHR ENCODING

Since our pre-training framework is compatible with most LLM structures, we investigated the impacts of using different LLMs as backbones for LLM4EHR. We compared the results obtained from small GPT-2 (Radford et al., 2019) against three other LLMs of comparable sizes: BERT (Devlin et al., 2019), Longformer (Beltagy et al., 2020) and RoBERTa (Liu et al., 2019). Additionally, we used versions of BERT (Alsentzer et al., 2019) and Longformer (Li et al., 2022) fine-tuned with MIMIC-III (Johnson et al., 2016) clinical notes. **Table 7** showed that our pre-training pipeline transfers well across different LLM structures, with GPT-2 (Radford et al., 2019) achieving more consistent results across all three tasks. In Appendix A.6, we show the relationship between this performance and the number of parameters the models contain. We attribute this observation to GPT-2's autoregressive nature, which is hypothesised to be more suitable for processing temporal data (Chang et al., 2025; Liu et al., 2024). Additional results using newer, larger LLMs are shown in **Section A.8**.

Table 7: The effects of different LLM backbones on the downstream task. We included LLMS of similar sizes in this evaluation (<200 million parameters) and highlighted the best results in bold.

| Tasks | Phenotyping | | Decompensation | | Remaining LoS | |
|---|---|---|---|---|---|---|
| Models | AU-ROC (macro) | AU-ROC (mirco) | AU-ROC | AU-PRC | RMSE | R2 |
| GPT-2 | **0.719 (0.003)** | **0.761 (0.006)** | 0.912 (0.003) | 0.333 (0.025) | 5.743 (0.088) | 0.129 (0.054) |
| RoBERTa | 0.708 (0.005) | 0.756 (0.006) | **0.916 (0.004)** | **0.413(0.012)** | 5.808 (0.102) | 0.109 (0.074) |
| BERT | 0.671 (0.004) | 0.749 (0.007) | 0.905 (0.004) | 0.378 (0.022) | 6.024 (0.072) | 0.032 (0.056) |
| Longformer | 0.710 (0.004) | 0.758 (0.004) | 0.912 (0.004) | 0.334 (0.013) | **5.681 (0.091)** | **0.147 (0.045)** |

THE CHOICE OF FINE-TUNING STRATEGIES

We evaluated the adaptability of LLM4EHR beyond linear probing. Here, we compare the performance of linear probing against end-to-end tuning, bottleneck adapters (Houlsby et al., 2019) and prefix tuning (Li & Liang, 2021). For end-to-end tuning, we used the same task heads as linear probing and unfroze LLM4EHR's parameters during tuning. We injected trainable adapters within the frozen attention blocks with bottleneck sizes of $64$, $96$ and $128$ as bottleneck adapters. Given that LLM4EHR relies on TS embedding for downstream predictions, we appended 20 trainable vectors to the TS embedding as a task-specific suffix. The dimensions of the task vectors were kept the same as the LLM4EHR's hidden dimension. Unlike bottleneck adapters, we report results only for prefix tuning with 20 trainable vectors, as we observed no significant performance gain with more prefix vectors. **Table 8** showed that LLM4EHR generalises well to different fine-tuning strategies. Additionally, **Table 8** showed that the impacts of downstream tuning strategies are task-specific, with end-to-end tuning demonstrating larger performance gain in classification tasks and bottleneck adapters demonstrating better performance in LoS regression.

## 6 DISCUSSION

We presented LLM4EHR as a multimodal clinical foundation model for modelling clinical time series observations and EHR sequences. The results in **Section 5** demonstrated that LLM4EHR

Table 8: The effects of different fine-tuning strategies on the downstream task. The numbers in the brackets indicate the sizes of the bottleneck adapters or the number of prefix tokens for prefix tuning; the best results are in bold.

| Tasks | Phenotyping | | Decompensation | | Remaining LoS | |
|---|---|---|---|---|---|---|
| Tuning | AUC-ROC (macro) | AUC-ROC (micro) | AUC-ROC | AUC-PRC | RMSE | R2 |
| Linear | 0.719 (0.003) | 0.761 (0.006) | 0.912 (0.003) | 0.333 (0.025) | 5.743 (0.088) | 0.129 (0.054) |
| End-to-end | **0.748 (0.005)** | **0.791 (0.006)** | **0.939 (0.004)** | **0.343 (0.015)** | 5.687 (0.072) | 0.130 (0.049) |
| Adapter (64) | 0.730 (0.002) | 0.773 (0.003) | 0.920 (0.004) | 0.336 (0.018) | 5.643 (0.074) | 0.154 (0.045) |
| Adapter (96) | 0.734 (0.003) | 0.777 (0.003) | 0.930 (0.003) | 0.339 (0.019) | 5.583 (0.069) | 0.161 (0.043) |
| Adapter (128) | 0.734 (0.002) | 0.777 (0.004) | 0.934 (0.003) | 0.341 (0.021) | **5.432 (0.071)** | **0.172 (0.045)** |
| Prefix (20 tokens) | 0.738 (0.004) | 0.781 (0.004) | 0.919 (0.004) | 0.335 (0.022) | 5.750 (0.080) | 0.128 (0.049) |

is generalisable to various downstream tasks and is transferable between EHR datasets. The interpretability study in **Section 5.4** showed that our $\omega$-regularised contrastive objective improves the consistency and interpretability of the learned TS embeddings. The major limitation of LLM4EHR is scalability to larger language models, which have considerable memory requirements to train. Future research could consider more efficient contrastive learning structures such as He et al. (2020), which uses a memory queue to store past samples for calculating the contrastive objective. The results in **Section 5.2** showed that LLM4EHR is less suitable for regression tasks than classification tasks, which warrants further investigation. Lastly, we evaluated the performance of LLM4EHR using learned TS embeddings due to labelled TS being omnipresent in all datasets. Future research should investigate the applicability of LLM4EHR for learning EHR event embeddings.

## ETHIC STATEMENT

The experiments in this manuscript are entirely data-driven, using anonymised EHR data, with no patient-identifiable information. We found no outstanding ethical or moral concerns with our methods. All datasets mentioned in our manuscript, apart from the private PICU dataset, are publicly available. The use of the private PICU dataset in our paper is approved by the healthcare provider and approved by an ethics committee. The source of the private PICU dataset is obscured for anonymisation and will be revealed in the print-ready copy of the manuscript, if applicable.

## REPRODUCIBILITY STATEMENT

All experiments carried out in **Section 5** can be reproduced by following the implementation detailed in **Section A.2**. The implementations of baseline models are detailed in **Section A.3**. All datasets and benchmarks, apart from the private PICU dataset, are publicly available. Finally, we made our code available in the supplementary material.

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

# A APPENDIX

## A.1 INFONCE LOSS

The InfoNCE loss (van den Oord et al., 2018), normalised as in He et al. (2020) and Kiyasseh et al. (2021) is defined as:

$$\mathcal{L}_{\text{NCE}} = -\sum_{i=1}^{P} \log \frac{\exp\left(\text{sim}(v_i', z_i')/\tau\right)}{\sum_{k \in P\setminus i} \exp\left(\text{sim}(v_i', z_k')/\tau\right)} \qquad \text{sim}(v_i', z_i') = \frac{v_i' \cdot z_i'}{\|v_i'\|\|z_i'\|} \qquad (5)$$

This loss objective is commonly used for contrastive learning. The temperature value $\tau$ is shared between **Equation 5** and **Equation 2** for calculating the alignment objective $\mathcal{L}_{\text{align}}$.

## A.2 DESCRIPTION OF DATASETS

We extracted 11 variables from the 4 datasets as TS features; these variables are: Diastolic blood pressure, Systolic blood pressure, Mean blood pressure, Fraction inspired oxygen, Heart rate, Temperature, pH, Respiratory rate, Oxygen saturation, Glucose and Glasgow coma scale total.

### A.2.1 MIMIC-III

Medical Information Mart for Intensive Care (MIMIC-III) is a large-scale, single-centre, intensive care EHR dataset (Johnson et al., 2016). For our work, we used labelled TS for 41902 ICU episodes extracted by Harutyunyan et al. (2019). 1061 episodes were removed due to having irregular mortality labels (death before admission to ICU) or having no corresponding EHR interactions, resulting in 40841 episodes for our experiments. We repartitioned the episodes with 28588 episodes for self-supervised pre-training, 8168 labelled episodes for fine-tuning and 4085 episodes for testing. We extracted corresponding EHR sequences for all ICU episodes.

Missing values in TS features were imputed via linear interpolation if other observations exist or normal values if no observations exist for the feature. The labels for in-unit mortality, phenotyping, decompensation and LoS regression were defined by Harutyunyan et al. (2019). The fine-tuning and testing episodes were not visible to the models during pre-training. We included lab observations, MetaVision (mv) procedures and intake (medication) events in EHR sequences. Note that ICD diagnosis events were not included as they were used to define phenotyping labels. For EHR events (lab and intake) associated numerical values, their numerical values were represented via special quantile tokens, same as in Renc et al. (2024).

### A.2.2 EICU

The eICU collaborative research database is a freely available multi-centre EHR dataset (Sheikhalishahi et al., 2020). Timestamps for EHR interactions and TS measurements are recorded in minutes relative to the start of ICU admission. Given the memory usage of processing the entirety of eICU, we used the first 30000 labelled TS extracted by Sheikhalishahi et al. (2020). Similar to **Section A.2.1**, we imputed missing observations in TS features via linear interpolation and normal values.

We further re-partitioned the dataset to be 21000 for self-supervised pre-training, 6000 for fine-tuning and 3000 for testing. The labels for in-unit mortality, phenotyping, decompensation and LoS regression were defined by Sheikhalishahi et al. (2020). All 30000 involved TS episodes had eligible EHR observations and labels. We extracted EHR sequences for the 30000 TS episodes using their lab and drug observations, as well as their treatment records. Events associated with numerical values were tokenised using the same method in **Section A.2.1**.

### A.2.3 PHYSIONET CHALLENGE 2012

The Physionet challenge 2012 (Silva et al., 2012) is an ICU dataset for predicting patient in-unit mortality using their first 48 hours of observations. The original dataset consists of 42 TS variables. To allow for transfer learning from MIMIC-III (Harutyunyan et al., 2019) and eICU (Sheikhalishahi et al., 2020), we only used 11 physiological variables that are common in these three datasets. Additionally, we resampled the patient observations hourly and used linear interpolation to impute

the missing observations. The Physionet 2012 dataset has 12000 samples in total, divided into three equal partitions of 4000 each. We used the competition settings for evaluating all the models, which involved training the models on set A and evaluating them on set C. The quality of the mortality predictions was measured in two custom metrics: the sensitivity score ($S_1$) and the $H$ score. The top results posted for the challenge were 0.5353 for the sensitivity score and 17.88 for the $H$ score. The sensitivity score is defined as:

| Outcome | | Observed | |
|---|---|---|---|
| | | Death | Survivor |
| Predicted | Death | TP | FP |
| | Survivor | FN | TN |

$$Se = TP/(TP + FN)$$
$$^+P = TP/(TP + FP)$$
$$S_1 = \min\left(Se, ^+P\right)$$

The $H$ score is based on the Hosmer-Lemeshow $H$ statistic. The in-unit mortality risks predicted by a model are first sorted and the corresponding records are binned into deciles designated by $g = 1, 2, 3 \cdots 10$, with each decile containing 400 samples. For each decile $g$, $O_g$ is the observed number of deaths, $E_g$ is the predicted number of deaths, $N_g$ is the number of records (400), and $\pi_g$ is the mean estimated risk for records in the decile. Therefore, the $H$ score is defined as:

$$D = \pi_{10} - \pi_1$$
$$H' = \sum_{g=1}^{10} \frac{(O_g - E_g)^2}{N_g \pi_g (1 - \pi_g) + 0.001}$$
$$H = H'/D$$

### A.2.4 PRIVATE PAEDIATRIC INTENSIVE CARE

The PICU dataset is a private EHR dataset consisting of 1300 unique paediatric ICU (PICU) stays at a paediatric hospital between May 01, 2019 to January 08, 2023. We only included PICU stays longer than 48 hours in our work. Similar to experiments on the Physionet Challenge 2012, we used the 11 shared TS variables for in-unit mortality predictions. The TS observations were resampled hourly and imputed via linear interpolation.

## A.3 IMPLEMENTATION OF BASELINE MODELS

All experiments were performed on an NVIDIA A100 GPU with 80 GB memory.

### A.3.1 GENERIC MODELS

Generic models include three deep learning structures commonly used for TS modelling: Multilayer perceptrons (MLP), Bidirectional Long Short-Term Memory Model (LSTM) and Convolutional Neural Networks (CNN). All models were implemented in PyTorch (Paszke et al., 2019).

The baseline MLP was implemented using two linear layers with ReLU activations. We applied the MLP model directly to the normalised TS features. We experimented with hidden sizes 32, 64, 128, 256, 512 and reported best results at hidden size 256. We trained the model for 10 epochs on all tasks using the Adam optimiser with a learning rate of 0.001. The training batch size was 64 for sequence classification and $64 \times 192$ for dynamic classification and regression.

The LSTM was implemented with two bidirectional LSTM layers and two linear layers. We performed a grid search over the number of LSTM layers (1 to 2) and hidden sizes (32, 64, 128, 256, 512). We reported the best results with two LSTM layers and a hidden size of 256. Similar to the experiments by Harutyunyan et al. (2019) and Sheikhalishahi et al. (2020), the LSTM was trained on the normalised TS features. The LSTM was trained for 10 epochs using the Adam optimiser with a learning rate of 0.001.

We adopted the 18-layer ResNet (He et al., 2016) as the CNN baseline. Given that CNN produces intermediate TS representations via convolution, it is not possible to obtain vector representations of individual time steps for dynamic classifications and regressions. The CNN baseline was trained with batch size 64 and a learning rate of 0.001 using the Adam optimiser. The operations of the CNN baseline are shown in **Table 9**.

Table 9: Implementation details for the CNN baseline.

| Layer name | Operation |
|---|---|
| conv1 | $8 \times 8$, 64, stride 1, padding 4 |
| conv2_x | $3 \times 3$ max pool, stride 2 |
| | $\begin{pmatrix} 3 \times 3 & 64 \\ 3 \times 3 & 64 \end{pmatrix} \times 2$ |
| conv3_x | $\begin{pmatrix} 3 \times 3 & 128 \\ 3 \times 3 & 128 \end{pmatrix} \times 2$ |
| conv4_x | $\begin{pmatrix} 3 \times 3 & 256 \\ 3 \times 3 & 256 \end{pmatrix} \times 2$ |
| conv5_x | $\begin{pmatrix} 3 \times 3 & 512 \\ 3 \times 3 & 512 \end{pmatrix} \times 2$ |
| | average pool, 1000d fc, softmax |

### A.3.2 TS MODELS

TS models include three self-supervised self-attention models for general TS modelling: SimMTM (Dong et al., 2023), PatchTST (Nie et al., 2023) and PatchTSMixer (Ekambaram et al., 2023). Inspired by prior research such as Liu et al. (2024) and Chang et al. (2025), we also included an ablated version of our pre-trained model, the supervised GPT-2 (Radford et al., 2019). The supervised GPT-2 is the version of LLM4EHR trained directly via task supervision, instead of using **Equation 4**.

During SimMTM pre-training, $n$ masked instances of a given TS are created. Masked copies of a TS are pulled closer via a contrastive objective. Similar to our work, the contrastive loss in SimMTM is defined with temperature $\tau$. We kept the size of SimMTM's transformer backbone consistent with the small GPT2 backbone used in LLM4EHR. Additionally, we performed grid search over model temperature $\tau = 0.02, 0.2, 1$ and masked ratio $r = 0.12, 0.25, 0.5, 0.75$. We reported the SimMTM fine-tuning results after pre-training with $\tau = 0.02$ and $r = 0.25$ as the combination produced best overall results in classifications. We kept $n = 3$ during pre-training as we observed a decline in model performance at $n = 4$.

We used the HuggingFace (Wolf et al., 2019) implementations for PatchTST (Nie et al., 2023) and PatchTSMixer (Ekambaram et al., 2023). We contextualise the pre-training task for both models as 48 hours forecasting. Here, the models are trained to forecast the next 48 hours of patient TS observations using the first 48 hours of TS observations as context. The pre-trained models are fine-tuned for downstream tasks, similar to our model. Both PatchTST and PatchTSMixer produce patched TS representations and we set the patch length to be 6 for both models. We reported the results for PatchTST with 3 encoder layers, 16 attention heads per layer and hidden size 128. We experimented with larger PatchTST model at 4 encoder layers with hidden size 256 and found no significant performance change. Given that we For PatchTSMixer, we perform grid search over the number of encoder blocks 2, 3, 4, 5, 6 and hidden sizes 32, 64, 128, 256, 512. We reported the best overall results for PatchTSMixer with 5 encoder blocks and hidden size 128.

### A.3.3 EHR MODELS

Event-Based Contrastive Learning (EBCL) by Jeong et al. (2024) is a transformer-based framework for aligning multimodal patient TS observation embeddings before and after key index events. The original experiments by Jeong et al. (2024) used ICU admissions as index events for contrastive pre-training and fine-tuned the model for downstream risk predictions. We implemented EBCL by following the official implementation and similarly used ICU admissions as index events. We note that we kept the hidden dimension of EBCL consistent with that of LLM4EHR at 768. As the original experiments were designed for sequence classifications, we only evaluated EBCL on in-unit mortality and phenotyping predictions.

King et al. (2023) provided a transformer-based framework for aligning patient TS embedding with clinical notes embedding. Their original experiments on MIMIC-III used a contrastive objective to align TS observations with clinical notes for a given ICU stay. Given the difficulty of re-extracting clinical notes for our experiments, we used their pre-trained model for in-unit mortality and phenotyping predictions. The pre-trained model was fine-tuned for downstream tasks using the same fine-tuning and testing sets as the other baseline models.

Similar to King et al. (2023), Wang et al. (2025) aligns patient clinical notes with TS observations using a contrastive objective. We note that their original implementation used a published pipeline for processing MIMIC-III notes by Khadanga et al. (2019) and did not include further processing to censor patient phenotyping labels from clinical notes (Khadanga et al. (2019) did not report results for patient phenotyping). Therefore, we reproduced CTPD using only nursing and progression notes extracted according to Khadanga et al. (2019), as implied by Figure 1 in CTPD's original publication. Different for the author's original implementation, we used BERT small instead of BERT tiny to be consistent with LLM4EHR's backbone, We experimented with different values for $\lambda_1 = 0.1, 0.5, 1, 2$ and $\lambda_2 = 0.1, 0.5, 1, 2$ and reported CTPD's results at $\lambda_1 = 0.5$ and $\lambda_2 = 0.5$ for best phenotyping performance.

Simply Attend and Diagnosis (SAnD) (Song et al., 2018) is a previously published self-attention model for modelling clinical time series that was evaluated on the same MIMIC-III benchmark as our experiment. In general, SAnD processes temporal data via multi-head self-attention layers and employs dense interpolation to produce sequence representations for classification. Since the original authors did not provide their implementations in Song et al. (2018), we re-implemented SAnD in Pytorch with autoregressive self-attention (Radford et al., 2019) and dense aggregation (Trask et al., 2015). We trained SAnD on the normalised TS features in both single-task and multi-task settings and were able to reproduce the original paper's reported results after hyperparameter tuning.

### A.3.4 OUR MODELS

We implemented LLM4EHR using open-sourced LLM weights hosted on HuggingFace (Wolf et al., 2019). The embedding dimensions for the TS encoding layers are the same as the token embedding dimensions of the LLM backbone. The models were pre-trained on MIMIC-III (Johnson et al., 2016) and eICU (Pollard et al., 2018) for 5 epochs with a learning rate of $0.0001$ and batch size of $64$. Due to reasons mentioned in **Section 5.4**, we kept the patch size to be $5$ time steps for all experiments. The value of the contrastive temperature $\tau$ is shared between the InfoNCE loss (**Equation 5**) and the $\omega$ weighted loss (**Equation 2**). The autoregressive decoding operation is defined as:

$$\{\hat{x}^2, \hat{x}^3, \cdots \hat{x}^T\} = \text{Decoder}(\{v_1, v_2, \cdots v_{T-1}\}) \tag{6}$$

$$\mathcal{L}_{\text{recon}} = \mathbb{E}_B[\frac{1}{T}\sum_{t=2}^{T}\|x_i^t - \hat{x_i}^t\|_2^2] \tag{7}$$

### A.4 ADDITIONAL RESULTS ON THE eICU DATASET

Similar to **Section 5.2**, we present the fine-tuning performance of LLM4EHR on the eICU (Pollard et al., 2018) dataset. We did not replicate the experiments by King et al. (2023) here due to reasons stated in **Section A.3.3**. The results from eICU (Pollard et al., 2018) are consistent with observations in **Section 5.2**, where LLM4EHR achieved improved accuracy in classification but underperformed against recent TS models for predicting Los.

Table 10: Evaluation metrics for all fine-tuning tasks in eICU Pollard et al. (2018), the results were averaged over ten runs and presented as mean (std), best results are indicated in bold.

| | | eICU | | | | | |
|---|---|---|---|---|---|---|---|
| Tasks | | Phenotyping | | Decompensation (hourly) | | Remaining LOS | |
| Models | | AUC-ROC (macro) | AUC-ROC (micro) | AUC-ROC | AUC-PRC | RMSE | R2 |
| Generic Models | MLP | 0.543(0.016) | 0.718 (0.003) | 0.745 (0.011) | 0.096 (0.006) | 2.63 (0.058) | -0.35 (0.005) |
| | CNN | 0.607 (0.026) | 0.753 (0.013) | N/A | N/A | N/A | N/A |
| | LSTM | 0.638 (0.003) | 0.769 (0.001) | 0.895 (0.003) | 0.267 (0.012) | 2.154 (0.060) | 0.305 (0.038) |
| TS Models | Supervised GPT | 0.690 (0.018) | 0.779 (0.016) | 0.867 (0.004) | 0.192 (0.017) | 2.048 (0.004) | 0.363 (0.009) |
| | SimMTM | 0.704 (0.002) | 0.791 (0.003) | 0.895 (0.003) | 0.267 (0.012) | **1.919 (0.011)** | **0.429 (0.009)** |
| | PatchTSMixer | 0.705 (0.004) | 0.782 (0.011) | 0.868 (0.003) | 0.177 (0.019) | 2.037 (0.009) | 0.357 (0.014) |
| | PatchTST | 0.686 (0.005) | 0.778 (0.006) | 0.867 (0.009) | 0.192 (0.010) | 2.040 (0.013) | 0.359 (0.024) |
| EHR Models | EBCL | 0.673 (0.011) | 0.763 (0.009) | N/A | N/A | N/A | N/A |
| | King et al. | N/A | N/A | N/A | N/A | N/A | N/A |
| | SAnD | 0.639 (0.017) | 0.732 (0.021) | 0.833 (0.006) | 0.178 (0.007) | 2.141 (0.004) | 0.314 (0.009) |
| Ours | LLM4EHR | **0.709 (0.008)** | **0.791 (0.015)** | **0.922 (0.007)** | **0.404 (0.015)** | 2.027 (0.004) | 0.336 (0.004) |

## A.5 THE INTERPRETABILITY ANALYSIS

The LoS regression results in **Section 5.2** indicated a trade-off between cross-modality alignment and retaining numerical features. We note that this observation is consistent with the reported results by Sheikhalishahi et al. (2020), where numerical TS features are better for predicting LoS. We performed K-means (Lloyd, 1982) clustering on patches of TS embedding, EHR embedding and normalised TS features and compared their cluster assignments as an interpretability analysis. We measured the Cross-Interpretability (CI) between TS and EHR embeddings by calculating the Adjusted Mutual Information (AMI) (Vinh et al., 2010) between their clusters. Similarly, we measured the Self-Interpretability (SI) between TS embedding and feature clusters. In this case, a high SI indicates that the learned TS embeddings are effective at preserving information from the original TS features. Conversely, a high CI indicates that the learned TS embeddings are effective at preserving information from EHR events. We determined the number of clusters for CI and SI by optimising the silhouette score (Rousseeuw, 1987) for EHR embedding and TS feature clusters. **Figure 4** shows the distortion of the numerical TS features in the embedding space, where training without $\mathcal{L}_\omega$ caused the learned TS embedding to form sparse micro-clusters.

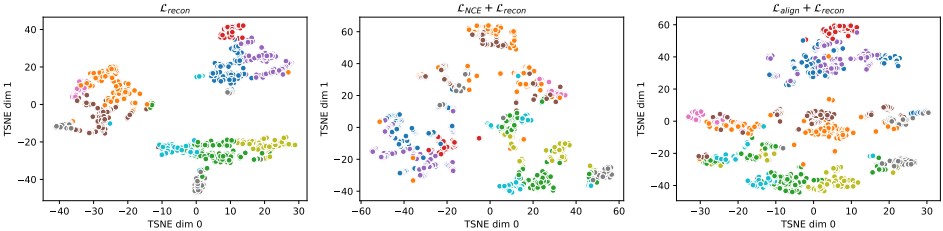

Figure 4: tSNE-projected (Hinton & Roweis, 2002) (perplexity=30) TS patch embeddings obtained from three different pre-training objectives; the data points are labelled using clustering on the raw numerical features.

**Table 11** shows the complete results for the interpretability analysis. Our pre-training objective achieved the highest CI between the EHR and TS embedding, with a $41\%$ improvement in CI after adding $\mathcal{L}_\omega$ to the overall objective. The best SI was achieved when training with the reconstruction objective only, which explains why LLM4EHR underperformed in LoS regression in **Section 5.2**. Additionally, there is a drop in CI when using both the InfoNCE and the reconstruction objective. We hypothesise that this is due to the trade-off between cross-modality aligning and preserving numerical features in our framework.

Table 11: CI, SI and Silhouette scores for clustering the TS embeddings obtained via different pre-training objectives.

| Loss | CI | SI | Silhouette |
|---|---|---|---|
| $\mathcal{L}_{\text{recon}}$ | 0.039 | **0.638** | **0.302** |
| $\mathcal{L}_{\text{NCE}}$ | 0.162 | 0.462 | 0.235 |
| $\mathcal{L}_{\text{NCE}} + \mathcal{L}_{\text{recon}}$ | 0.146 | 0.561 | 0.236 |
| $\mathcal{L}_{\text{NCE}} + \mathcal{L}_{\omega} + \mathcal{L}_{\text{recon}}$ | **0.208** | 0.575 | 0.163 |

## A.6 THE RELATIONSHIP BETWEEN PERFORMANCE AND THE NUMBER OF LLM PARAMETERS

In **Figure 5** we present the results of two downstream tasks (given in **Table 7**) against the number of parameters the models contain.

This illustrates that there is no clear relationship between the LLM model size and the final model performance. However in this case, as well as containing different numbers of parameters, models are architecturally different and were originally trained on different datasets by different studies. If the deployment setting is strictly limited by compute and memory, BERT could provide an option. However, it comes at the cost of reduced performance on downstream tasks compared to our recommended LLM, GPT-2, which strikes a better balance between performance and efficiency.

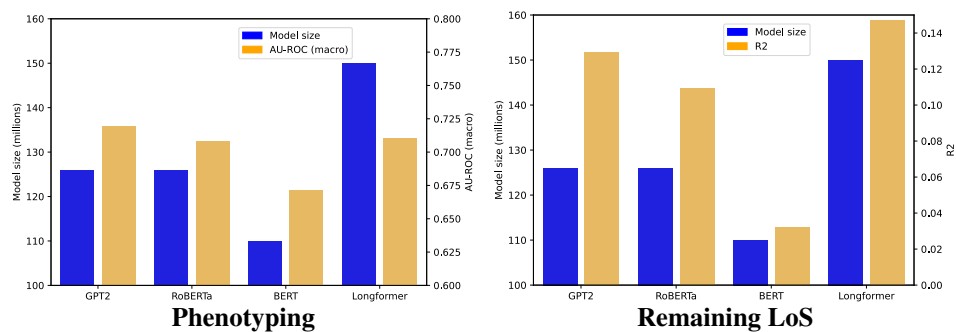

Figure 5: The performance of different LLMs on phenotyping (macro AU-ROC) and remaining LoS regression (R2) against the number of parameters

## A.7 SCALABILITY

Continuing from **Section A.6**, we experimented with different versions of GPT-2 (Radford et al., 2019). We could only train GPT-XL (1.5B parameters) using a batch size of 16 due to the memory usage issue discussed in **Section 6**. Similar to **Section A.6**, **Table 12** demonstrates no clear relationship between LLM size and fine-tuning performance. Therefore, we conclude that the underlying architecture of LLMs has a greater impact on performance than their size when applied to LLM4EHR.

Table 12: The scalability of LLM4EHR with GPT-2 backbone, all models were pre-trained with a batch size of 16, best results are indicated in bold.

| Task | Phenotyping | | Decompensation | | Remaining LoS | |
|------|-------------|--|----------------|--|---------------|--|
| Size | AUC-ROC (macro) | AUC-ROC (micro) | AUC-ROC | AUC-PRC | RMSE | R2 |
| 124M | 0.692 (0.003) | 0.745 (0.005) | 0.910 (0.003) | 0.328 (0.017) | 5.731 (0.026) | 0.143 (0.014) |
| 355M | **0.707 (0.003)** | **0.757 (0.006)** | 0.900 (0.004) | 0.319 (0.021) | 5.699 (0.021) | 0.162 (0.013) |
| 744M | 0.699 (0.004) | 0.750 (0.005) | 0.909 (0.003) | **0.330 (0.018)** | 5.702 (0.030) | 0.162 (0.012) |
| 1.5B | 0.706 (0.004) | 0.755 (0.008) | **0.912 (0.003)** | 0.329 (0.021) | **5.682 (0.031)** | **0.184 (0.014)** |

## A.8 ADDITIONAL RESULTS WITH AUTOREGRESSIVE LLMS

We investigated the impacts of using different LLMs as backbones for LLM4EHR in **Section 5.4**, with GPT-2 (Radford et al., 2019) demonstrating more consistent results across all three tasks. In **Section 5.4**, we limited our comparison to LLMs comparable in size to small GPT-2. Here, we present additional results obtained from Llama 3.2 (Touvron et al., 2023) (1 billion parameters) and OPT (Zhang et al., 2022) (1.3 billion parameters). The results are compared with those obtained from GPT-2 XL (Radford et al., 2019) (1.5 billion parameters) in **Section A.7**. **Table 13** shows that Llama outperformed GPT-2 XL in all tasks, despite having fewer parameters. OPT performed on par with GPT-2 XL in classifications but achieved the best performance in LoS regression. Combined with the results in **Section A.7**, the additional results show that the scaling behaviour of LLM4EHR is influenced by both the structure and the size of the LLM backbone.

Table 13: Additional results with Llama 3.2 and OPT, compared against GPT-XL, all models were pre-trained with a batch size of 16, best results are indicated in bold.

| Task | Phenotyping | | Decompensation | | Remaining LOS | |
|------|-------------|--|----------------|--|---------------|--|
| Model | AUC-ROC (macro) | AUC-ROC (micro) | AUC-ROC | AUC-PRC | RMSE | R2 |
| OPT | 0.705 (0.003) | 0.756 (0.005) | 0.914 (0.004) | 0.329 (0.011) | **5.677 (0.025)** | **0.187 (0.018)** |
| Llama 3 | **0.714 (0.006)** | **0.758 (0.007)** | **0.915 (0.004)** | **0.330 (0.008)** | 5.680 (0.030) | 0.184 (0.011) |
| GPT-2 XL | 0.706 (0.004) | 0.755 (0.008) | 0.912 (0.003) | 0.329 (0.021) | 5.682 (0.031) | 0.184 (0.014) |

## A.9 EFFECTS OF PRE-TRAINING BATCH SIZE

We experimented with different batch sizes during pre-training. As shown in **Table 14**, using a larger batch size during pre-training improved the model's performance in downstream classification tasks. In contrast, the remaining Los prediction benefitted from pre-training with a small batch size. Specifically, we observed a noticeable decline in LoS regression accuracy when the pre-training batch size increased from 16 to 32.

Table 14: The effects of pre-training batch size, best results are highlighted in bold.

| Task | Phenotyping | | Decompensation | | Remaining LoS | |
|------|-------------|--|----------------|--|---------------|--|
| $B$ | AUC-ROC (macro) | AUC-ROC (micro) | AUC-ROC | AUC-PRC | RMSE | R2 |
| $B = 16$ | 0.692 (0.003) | 0.745 (0.005) | 0.910 (0.003) | 0.328 (0.017) | **5.731 (0.026)** | **0.143 (0.014)** |
| $B = 32$ | 0.704 (0.001) | 0.753 (0.005) | 0.911 (0.005) | 0.334 (0.017) | 5.798 (0.157) | 0.124 (0.047) |
| $B = 64$ | **0.719 (0.003)** | **0.761 (0.006)** | 0.912 (0.003) | 0.333 (0.025) | 5.743 (0.088) | 0.129 (0.054) |
| $B = 128$ | 0.714 (0.003) | 0.758 (0.006) | **0.917 (0.004)** | **0.346 (0.018)** | 5.789 (0.109) | 0.126 (0.032) |

## A.10 ADDITIONAL FEW-SHOT EXPERIMENTS

Following the few-shot experiments in **Table 4**, we conducted additional fine-tuning experiments using 10% and 5% of the labelled data. **Table 15** and **Table 16** show that LLM4EHR's consistently outperformed baseline models in all data partitions, with significant gains in classification tasks.

Table 15: Evaluation metrics for all fine-tuning tasks in MIMIC-III (Johnson et al., 2016) at 10% training data, the results were averaged over ten runs and presented as mean (std), best results are indicated in bold.

| | | MIMIC-III (10% training data) | | | | | |
|---|---|---|---|---|---|---|---|
| Tasks | | Phenotyping | | Decompensation (hourly) | | Remaining LoS | |
| Models | | AUC-ROC (macro) | AUC-ROC (micro) | AUC-ROC | AUC-PRC | RMSE | R2 |
| Generic Models | MLP | 0.550 (0.025) | 0.620 (0.029) | 0.784 (0.099) | 0.101 (0.062) | 7.771 (0.055) | -0.260 (0.030) |
| | CNN | 0.606 (0.012) | 0.657 (0.008) | N/A | N/A | N/A | N/A |
| | LSTM | 0.590 (0.014) | 0.659 (0.009) | 0.865 (0.041) | 0.301 (0.023) | 6.417 (0.101) | 0.077 (0.042) |
| TS Models | Supervised GPT | 0.584 (0.009) | 0.643 (0.019) | 0.844 (0.015) | 0.229 (0.025) | 6.498 (0.092) | 0.044 (0.095) |
| | SimMTM | 0.638 (0.008) | 0.680 (0.035) | 0.866 (0.018) | 0.317 (0.046) | 6.137 (0.088) | **0.149 (0.065)** |
| | PatchTSMixer | 0.619 (0.010) | 0.662 (0.005) | 0.868 (0.012) | 0.320 (0.026) | 6.275 (0.084) | 0.051 (0.055) |
| | PatchTST | 0.628 (0.008) | 0.663 (0.005) | 0.863 (0.021) | 0.273 (0.037) | 6.140 (0.092) | 0.129 (0.048) |
| EHR Models | EBCL | 0.630 (0.007) | 0.668 (0.012) | N/A | N/A | N/A | N/A |
| | King et al. | 0.646 (0.010) | 0.691 (0.006) | N/A | N/A | N/A | N/A |
| | CTPD | 0.630 (0.009) | 0.694 (0.005) | N/A | N/A | N/A | N/A |
| | SAnD | 0.600 (0.014) | 0.664 (0.017) | 0.846 (0.028) | 0.292 (0.037) | 6.147 (0.075) | 0.073 (0.079) |
| Ours | LLM4EHR | **0.664 (0.009)** | **0.702 (0.012)** | **0.885 (0.006)** | **0.326 (0.026)** | **6.006 (0.099)** | 0.125 (0.067) |

Table 16: Evaluation metrics for all fine-tuning tasks in MIMIC-III (Johnson et al., 2016) at 5% training data, the results were averaged over ten runs and presented as mean (std), best results are indicated in bold.

| | | MIMIC-III (5% training data) | | | | | |
|---|---|---|---|---|---|---|---|
| Tasks | | Phenotyping | | Decompensation (hourly) | | Remaining LoS | |
| Models | | AUC-ROC (macro) | AUC-ROC (micro) | AUC-ROC | AUC-PRC | RMSE | R2 |
| Generic Models | MLP | 0.380 (0.022) | 0.424 (0.032) | 0.592 (0.090) | 0.080 (0.058) | 10.885 (0.070) | -0.479 (0.028) |
| | CNN | 0.456 (0.019) | 0.467 (0.010) | N/A | N/A | N/A | N/A |
| | LSTM | 0.463 (0.016) | 0.484 (0.011) | 0.716 (0.045) | 0.244 (0.025) | 9.204 (0.110) | -0.142 (0.039) |
| TS Models | Supervised GPT | 0.461 (0.010) | 0.472 (0.019) | 0.716 (0.020) | 0.186 (0.027) | 8.899 (0.078) | -0.154 (0.195) |
| | SimMTM | 0.524 (0.012) | 0.503 (0.038) | 0.745 (0.025) | 0.267 (0.047) | **8.293 (0.092)** | -0.033 (0.077) |
| | PatchTSMixer | 0.511 (0.010) | 0.490 (0.007) | 0.749 (0.022) | 0.274 (0.029) | 9.080 (0.080) | -0.108 (0.063) |
| | PatchTST | 0.533 (0.009) | 0.491 (0.008) | 0.745 (0.022) | 0.235 (0.040) | 8.796 (0.090) | -0.054 (0.054) |
| EHR Models | EBCL | 0.535 (0.009) | 0.508 (0.015) | N/A | N/A | N/A | N/A |
| | King et al. | 0.554 (0.013) | 0.552 (0.012) | N/A | N/A | N/A | N/A |
| | CTPD | 0.542 (0.012) | 0.570 (0.011) | N/A | N/A | N/A | N/A |
| | SAnD | 0.521 (0.015) | 0.551 (0.021) | 0.740 (0.030) | 0.259 (0.040) | 8.697 (0.078) | -0.073 (0.082) |
| Ours | LLM4EHR | **0.577 (0.010)** | **0.588 (0.014)** | **0.778 (0.012)** | **0.293 (0.028)** | 8.293 (0.090) | **-0.027 (0.070)** |

