# OpenReview forum: "LLM4EHR: Aligning Clinical Time Series with Medical Event Sequences via Large Language Models"
_ICLR.cc/2026/Conference — Submitted to ICLR 2026_

### Official Review · Reviewer_yyS2 · 2025-10-27

**Soundness:** 4
**Presentation:** 4
**Contribution:** 3
**Rating:** 8
**Confidence:** 3

**Summary:**

​​LLM4EHR is a clinical foundation model that temporally aligns ICU time-series measurements with EHR event sequences using a frozen LLM encoder, patch-wise pooling, and a regularized contrastive objective to learn shared cross-modal representations. In experiments on mimic-iii and eICU, these embeddings improved downstream task performance and transferred to new cohorts with minimal performance loss.

**Strengths:**

- Clear, principled cross-modal alignment using an EHR-similarity-weighted contrastive objective that mitigates class collision.
- The problem being solved matters, EHR event sequences and physiologic time series are usually modeled separately, losing crucial temporal context that impacts diagnosis, risk prediction, and treatment timing, aligning them can boost accuracy and generalization with fewer labels.
- Practical temporal handling via non-overlapping time patches that bridge sparse EHR events and dense time-series data.
- Stable use of a frozen LLM with only new clinical token embeddings trained, leveraging general semantics without drift.
- Auxiliary autoregressive reconstruction preserves numeric fidelity of physiologic signals, with ablations showing its value.
- Strong, consistent improvements across multiple ICU prediction tasks and solid cross-dataset transferability.

**Weaknesses:**

- The few-shot claim is weakly supported; the experiments fine-tune on relatively large labeled cohorts and don’t show behavior at truly low-label regimes.
- Length-of-stay performance lags specialized baselines, and the paper offers limited concrete strategies to mitigate this gap.
- Heavy reliance on frozen LLM semantics for many new clinical tokens is unvalidated, there's no check that these learned token embeddings are clinically coherent.
- The choice to normalize target-domain data with source statistics is not standard and could bias transfer results, no ablation compares against target-stats normalization.

**Questions:**

- Why normalize target cohorts with source means/variance, can you provide an ablation with target stats normalization?
- How are patches with no EHR events handled in the alignment losses and what are the gradient implications?
-  How did you or would you validate the semantic quality of new clinical token embeddings, and would light adapter-tuning help?
- I wonder how sensitive are results to patch size and do you support off-diagonal alignment to capture realistic delays between orders, administrations, and physiologic response?
- Can you show learning curves for 1–5–10% labeled data or k-shot per phenotype to substantiate the few-shot claim?
- Could you report calibration metrics (brier) and whether the EHR-weighted term improves or harms calibration versus the baseline objective?
Minor:
- in line 67, duplicated word: calculating
- line 169-170, word predictive overused
- line 346, missing rationale, why the sequences were truncated at 200h?
- l. 486 should be ETHICS
- Please fix table headers with mirco instead of macro, and marco -> macro

---

> ### Author Response · Authors · 2025-11-21
> **Response to the reviewer**
>
> We appreciate the balanced feedback from the reviewer. We address the significant questions and list planned additions to our manuscript in our point-by-point response below. Additionally, we thank the reviewer for pointing out spelling/style errors in the manuscript, which will be corrected in the final version.
>
> ### Regarding the few-shot evaluation scenario:
> While we argued in Section 5.1 that the few-shot evaluation was more realistic for a real-world deployment setting and made no claims regarding LLM4EHR’s performance in an extremely low-data setting, we find the reviewer’s suggestions for adding additional few-shot scenarios reasonable. We are actively preparing new experiments with few-shot scenarios using 1%, 5% and 10% of the fine-tuning partition as suggested by the reviewer.
>
> ### Regarding the Length-of-stay prediction performance:
> We agree with the reviewer’s observation that LoS results lag specialised TS baselines (In Section 5.2/A.5). Without making additional claims beyond the scope of our manuscript, we acknowledge that LoS regression is one of the most challenging tasks in general clinical outcome predictions. A common (and arguably more realistic) strategy is to aggregate LoS into bins, as noted by multiple papers we cited in our manuscript, such as Harutyunyan et al. (2019) and Song et al. (2019). We decided against repurposing LoS regression into a multilabel classification task, as further analysis in Section A.5 demonstrated an interesting trade-off between maximising cross-modal alignment and reducing the self-interpretability of the learned numerical TS feature embedding.
>
> ### Regarding the concern of the quality of EHR token embedding:
> In Section 4, we specified that we kept the LLM’s attention/embedding weights frozen to keep $z_{i}$ stable. Since $\omega$ is calculated directly using the cosine similarity between EHR patch embeddings (Section 4 and Section A.1), we provided an ablation study in Section 5.4 where we showed that adding the $\mathcal{L}_{\omega}$ term improved the LLM4EHR’s performance, validating our choice of employing a frozen LLM. However, we do acknowledge that similarity-based token retrieval was involved in some of our cited works (Renc et al., 2024; Kraljevic et al., 2024), which we believed would be best left for future works since we did not involve EHR token interpretation as part of our original contribution (Section 6).
>
> ### Regarding the normalisation:
> We normalised the data using source statistics based on the assumption that a foundation model, trained on a large dataset, would be deployed to a specialised hospital using new patient data. In this case, perhaps more obvious to the PICU dataset (exclusively children) than the Physionet challenge dataset, we don’t expect to know the data statistics of the target dataset. The goal of transfer learning is, therefore, to investigate if pre-trained LLM4EHR would be adaptive to new data with fine-tuning. However, we acknowledge that we didn’t provide statistics for the four datasets or comparison results against models directly trained on the target datasets. We will address this for the final version of our manuscript.
>
> ### Regarding the specific questions:
> **Q1:** See above.
>
> **Q2:** We understand that the padding process was not well-explained in our paper, and we’ll amend the manuscript accordingly. Empty EHR patches would produce an aggregated embedding from padding tokens. This is intentional so we could avoid computationally expensive patch-wise operations during training. Empty patches would contribute to the contrastive SoftMax but carry low semantic similarities to other EHR patches. As gradients only propagate through TS embedding/decoding layers, empty EHR patches would yield weaker alignment gradients for their corresponding TS patches.
>
> **Q3:** See above.
>
> **Q4:** We acknowledged in Section 5.4 that we chose the patch size (a 5-hour window) to avoid excessive sparsity in EHR patches, which is important, as noted in Section 4, the timing of EHR events is usually noisy, depending on the exact EHR system. Off-diagonal alignment is possible by replacing L nce with a temporally scaled variant of contrastive loss, such as that shown in Lee et al. 2024 (cited in Section 4). We opted against this as 1) we already relaxed the temporal alignment between EHR events and TS observations by aggregation, and 2) we focused on similarity-based weighting between patches as our original contribution in the paper.
>
> **Q5:** We plan to expand the evaluation with fine-tuning using 1–5–10% labelled data as mentioned above.
>
> **Q6:** Given that calibration tests were uncommon in our cited baselines and benchmarks, we only reported calibration metrics on the Physionet Challenge 2012 dataset (H-metric was based on Hosmer-Lemeshow H statistic, see A.2.3), to remain consistent with the original setting of the challenge.
>
> **Minor:** Will be addressed during editing.

---

### Official Review · Reviewer_qYea · 2025-10-31

**Soundness:** 3
**Presentation:** 3
**Contribution:** 3
**Rating:** 4
**Confidence:** 4

**Summary:**

The paper introduces LLM4EHR, a multimodal clinical foundation model that jointly learns from electronic health record event sequences and clinical time series data. The authors argue that prior approaches fail to capture the temporal dependencies between these two modalities and propose a contrastive alignment objective that aligns TS observations with EHR event embeddings in a shared latent space. The model leverages pre-trained language model embeddings to regularize the contrastive loss and reduce class collision during training. LLM4EHR is trained on MIMIC-III and eICU datasets and evaluated across several downstream prediction tasks such as mortality, phenotyping, decompensation, and length of stay. Results show improved few-shot and cross-dataset generalization compared to baseline models. The paper concludes that the approach improves interpretability and transferability but faces scalability limitations due to memory demands of large LLMs.

**Strengths:**

Below are the strengths of the paper in my opinion:
1. Proposes a clear methodological contribution for temporal contrastive alignment using LLM embeddings.
2. Demonstrates consistent performance improvements across multiple clinical prediction benchmarks (mortality, phenotyping, decompensation, and length of stay) under few-shot and cross-dataset settings.
3. Includes interpretability analysis showing improved consistency of learned embeddings.
4. Evaluates on diverse datasets (MIMIC-III, eICU, Physio2012, and a private PICU dataset) with transparent experimental setup.

**Weaknesses:**

The major weaknesses of the paper in my opinion follows:

1. Offers an incremental contribution that primarily combines established contrastive and LLM-based methods rather than introducing a new paradigm.
2. The "foundation model" claim is overstated given the limited dataset scale and scope of downstream tasks.
3. Experimental analysis lacks depth (few ablations, no statistical significance reporting, limited robustness discussion).
4. Scalability remains a limitation due to high computational requirements for larger LLMs. (Although it is already noted in the paper)
5. Minimal qualitative or clinical validation beyond numerical benchmarks.
6. There is no clear comparison against recent multimodal architectures that jointly model structured and unstructured EHR data (e.g., transformer-based fusion models).
7. The current interpretability for such a clinical task lacks rigour and can be substantially improved.

**Questions:**

1. Please provide precise LLM configuration and compute profile. Please specify the exact sequence length, tokenizer choices, and any truncation rules for EHR events or TS tokens. Also report pretraining steps, batch sizes, device count, total GPU hours, and peak memory. This will help assess scalability, which you list as a key limitation.
2. You reformulate the commonly used contrastive objective to temporally align TS observations with EHR event sequences. Please formalize the positive and negative pair construction in time, the windowing or lag structure, and how you handle irregular sampling or missing TS. I think adding a figure or pseudo-code that shows how pairs are built over time would greatly improve this part.
3. Tables show means with parentheses that are std. Please add significance tests for key claims (few-shot gains, cross-dataset transfer.
(minor comment: For all tables please specify what the numbers in parentheses are (i.e., std).)
4. You repartition to 70% self-supervised pretraining, 20% fine-tuning, 10% testing. I suspect you did this already but can you please confirm splits are at the patient level and that no patient overlap exists across partitions or datasets (especially since for these datasets a given patient might have multiple visits so the visit id defers but patient id is the same). Describe any harmonization across MIMIC-III, eICU, and PhysioNet to avoid label or feature leakage, especially for remaining length of stay.
5. You state that, inspired by prior work, the model can make dynamic downstream predictions, such as an hourly mortality forecast. Yet the evaluation emphasizes classification. Can you please provide a forecasting setup with proper rolling origin evaluation, and reconcile this with the later statement that the method is less suitable for regression.
6. Please enumerate baseline implementations and hyperparameter search spaces in the main text or appendix. Clarify whether recent multimodal fusion baselines were included, not just generic TS or EHR models, since your contribution is cross-modality alignment.
7. I think adding calibration and other clinically meaningful decision metrics where applicable improves the paper. This will make the cross-dataset claims stronger, especially for mortality risk.
8. For reproducibility, how will the PICU-dependent steps be handled. (I assume that private data is not going to be released).
9. Can you please define how continuous TS vectors map to tokens, the discretization or projection used, and how variable-length episodes are handled?
10. You note poorer regression performance. Please provide a short analysis of failure modes and whether alternative reconstruction losses, discretized targets, or ordinal objectives improved results.
11. I think the description for the exact few-shot protocol can be improved by providing much more detail in the appendix such as shots per class, selection strategy, number of repeats, and how hyperparameters were tuned without peeking. This is important for interpreting "few-shot" gains.
Evidence: “Few-shot evaluation on MIMIC-III” in the outline.

---

> ### Author Response · Authors · 2025-11-21
> **Response to the reviewer**
>
> We thank the reviewer for the constructive and balanced review to our paper. Here is our pint-by-point responses to the reviewer’s questions:
>
> ### Regarding contribution:
> We believe that our intended scope of contribution is explained in Section 2. We also point out that multimodal alignment using EHR data with cross-dataset evaluation across four datasets is uncommon in the existing literature, which we believe is sufficient to demonstrate our contribution to existing works.
>
> ### Regarding the claim of “foundation models”:
> The definition of foundation models in EHR and clinical data modelling depends on the intended application. Prior works involving pre-trained tokenised EHR sequence models are often referred to as foundation models, even if their evaluation focuses exclusively on general outcomes predictions (See Wornow et al., 2023. In our cited papers).
>
> ### Regarding comparison against transformer fusion models:
> Comparisons against transformer/transformer fusion models (King et al) were provided in Section 5.1. However, we acknowledged that we primarily focused on comparisons with TS models. We limited comparisons to text/ts fusion models because they are less relevant to the dynamic data alignment objective we proposed (Section 2).
>
>
> ### Regarding the questions:
>
> **Q1:** Implementation details are provided in Appendix A.3, Section 5.4, and Tables 7, 11, and 12. We plan to organise the hardware details (GPU configuration, GPU hours, and peak memory) in a single table in the appendix. Please note that the exact GPU we used during our experiments are already specified in Appendix A.3.
>
> **Q2:** The patch aggregation/alignment process is described in Figure 3, with Figure 3a describing patch aggregation and Figure 3b describing the definition of positive pairs. Irregular sampling of EHR events and imputing missing TS values are addressed in Section A.2. We will move relevant sections closer to Figure 3 during editing. Please see our response to Reviewer yyS2, Q2, on how we handled empty EHR patches and their effects on the gradients, which will also be added to the paper.
>
> **Q3:** We’ll change some of the table descriptions to be more descriptive. Statistically significant results (P<0.05) were highlighted in bold, we will specify this during editing. Additionally, we provided an H-score for results conducted in the Physionet2012 challenge datasets, which was the model calibration metric used by the challenge.
>
> **Q4:** We can confirm no patient overlapping between data partitions and will explicitly state that in the appendix. Patient-level partition was intended by the authors of the MIMIC-III and eICU benchmarks we included in our paper. The dataset harmonisation process is in section A.2, which describes the common TS variables we used for all datasets. Remaining LoS is a rolling prediction made hourly, and we didn’t include any administrative status codes (discharge, death, transfer to end-of-life care, etc) and diagnostic codes (ICD 10, Apache IV etc), which could leak information regarding patient discharge and phenotype.
>
> **Q5:** Both decompensation and remaining LoS regression are rolling, hourly predictions as explicitly stated in Section 5.2. The definition of rolling prediction tasks was consistent with the benchmarks Harutyunyan et al. (2019) and Sheikhalishahi et al. (2020)., which we did not modify.
>
> **Q6:** We’ll include additional implementation details for the baseline models in appendix. Summary of multimodal EHR models are shown in Section 5.1.
>
> **Q7:** Given that calibration tests were uncommon in our cited baselines and benchmarks, we only reported calibration metrics on the Physionet Challenge 2012 dataset (H-metric was based on Hosmer-Lemeshow H statistic, see A.2.3), to remain consistent with the original setting of the challenge.
>
> **Q8:** Since we can’t publish the PICU dataset, we included a mechanically similar dataset (Physionet2012), which includes all the steps necessary to reproduce our experiments on the PICU dataset. The stats for all four datasets will be added in the revision.
>
> **Q9:** The process of how TS observations are embedded and processed by a frozen LLM was explained in the background section, as similar methods have been previously shown in our cited materials (such as Liu et al. (2024) from the related work section). We focused on our original contribution in the method section. A more detailed summary of our implementation will be added to the appendix during editing, with appropriate anchors in the main text.
>
> **Q10:** We refer the reviewer to Section 5.4 (ablation with different pre-training objectives) and Section A.5 (interpretability analysis with different pre-training objectives).
>
> **Q11:** Since we planned to expand more few-shot scenarios following other reviewers’ feedback, we’ll change the relevant descriptions accordingly after new results are collected.

---

### Official Review · Reviewer_Xd8N · 2025-11-05

**Soundness:** 3
**Presentation:** 3
**Contribution:** 3
**Rating:** 6
**Confidence:** 5

**Summary:**

The paper proposes LLM4EHR, a clinical foundation model that temporally aligns ICU EHR event sequences with numerical clinical time series (TS) to learn cross-modal patient representations. The method freezes a pre-trained LLM backbone and builds patch-level embeddings for EHR and TS, then trains with a regularized contrastive objective to align modalities, plus a TS reconstruction loss. Experiments on MIMIC-III and eICU, with transfer to PhysioNet 2012 and a paediatric PICU cohort, show consistent gains on classification tasks (phenotyping, decompensation, mortality) and competitive but not SOTA performance on remaining length of stay (LoS) regression. Ablations cover temperature τ, patch size (five-hour windows), and LLM backbone choices.

**Strengths:**

1.Originality: Introduces semantic-weighted contrastive alignment between EHR and TS at the temporal patch level, which is a meaningful extension of multi-modal contrastive learning in clinical settings.

2.Quality: Broad evaluation (few-shot hints, in-domain and cross-dataset) and ablations (τ, patch size/backbone). Cross-dataset mortality results show robust transfer.

3.Clarity: The training objective and data flow are well presented (overview figure, patching diagram, tables).

**Weaknesses:**

1.Regression performance / numerical fidelity: Remaining LoS performance is only competitive; the paper itself hypothesizes TS embedding distortion. Consider adding variable-level numeric reconstruction, distribution/quantile losses, or hierarchical multi-tasking to improve numerical fidelity and report the impact on LoS.

2.Robustness of semantic weighting (ω): If EHR coding is sparse/noisy or mismatched across sites/ages, ω could mislead alignment. Please simulate label/semantic noise, compare against unweighted or asymmetric weighting schemes, and quantify degradation.

3.Temporal alignment granularity: Fixed, non-overlapping five-hour patches may miss asynchronous or delayed effects common in ICU. Explore adaptive/learned patching, overlapping windows, or soft DTW-like temporal weights.

**Questions:**

Q1 (critical): How robust is ω under coding-system changes or age-group shifts (adult ↔ paediatric)? Please report cross-site/cross-coding ablations or controlled noise experiments (e.g., token description perturbation, increased OOV rate).
Q2: For LoS, does adding variable-level numeric reconstruction or quantile losses improve RMSE/R² without hurting classification? A small ablation in the appendix would help.

---

> ### Author Response · Authors · 2025-11-21
> **Response to the reviewer**
>
> We thank the reviewer for their balanced and detailed feedback on our paper. We especially thank the reviewer for suggesting alternative reconstruction objectives, which could help us expand the discussion in our paper regarding possible future works. Here are our point-by-point responses to the reviewer’s questions:
>
> ### On the impact of different reconstruction objectives:
> We agree with the reviewer that the TS reconstruction objective is perhaps more impactful on LoS regression than other classification tasks. Since we focused on contrastive alignment as our original contribution, we used a generic reconstruction objective as the auxiliary objective to isolate the performance impacts of adding our contrastive objective. Given the scale and complexity of the experiments we presented, we avoided experimenting with more complex reconstruction objectives, as they are not central to our original intended contribution.
>
> ### On the robustness of $\omega$:
> The sparsity and noisiness of EHR coding were similarly acknowledged in our paper as one of the most serious challenges to at-scale modelling of EHR data. We included cross-dataset experiments across two large ICU datasets with different coding systems (MIMIC/eICU), providing evidence that LLM4EHR is reasonably robust to shifts in coding schemes. Additionally, since the four datasets we used were collected from four different institutions (eICU and Physionet 2012 were multi-centre), the high level of uncertainty resulting from disparate data collection/reporting standards made designing controlled noise simulations especially challenging.
>
> ### On the prospect of dynamic temporal alignment:
> The reviewer’s suggestion is similar to another reviewer’s suggestion for adding off-diagonal alignment between patches (see Figure 3b). Such alignment is possible by replacing the auxiliary reconstruction objective with a weighted self-alignment objective (See Lee et al. 2024 in our cited papers). However, as mentioned in our paper, the choice of reconstruction objective was not part of our original contribution, which would be best addressed in a separate paper. We’ll amend the discussion section accordingly.
>
> ### Response to specific questions:
> **Q1:** Cross-dataset evaluation between MIMIC-III and eICU serves as a proxy for coding system change as these two systems employ different coding standards. The consistent results across these two EHR systems suggests that LLM4EHR is reasonably robust to changes in coding systems.
>
> **Q2:** We kept the auxiliary reconstruction objective generic to isolate the effect of our contrastive alignment objective.

---

### Official Review · Reviewer_2Zsg · 2025-11-07

**Soundness:** 2
**Presentation:** 2
**Contribution:** 3
**Rating:** 4
**Confidence:** 4

**Summary:**

The author proposes LLM4EHR on general ICU data. LLM4EHR is built on pretrained LLMs to embed unstructured EHR text data, as well as autoregressively recover time series values from their latent representation. The major contribution in LLM4EHR is the temporally aligned embedding of EHR text records and time series records, as well as an additional regularization loss term to address the issue of class collision. Empirically, the author demonstrates that the embeddings from LLM4EHR improve various downstream tasks.

**Strengths:**

* The goal of this work, which is to improve the analysis of ICU data using both time series data and EHR data, has a significant impact and is beneficial to healthcare research.
 * The major novelty in the proposed work, aligning the embeddings of two modalities for a feasible contrastive loss, and an additional regularization loss utilizing the feature of LLM, is reasonable.
 * The experiments cover a variety of downstream tasks, showing both strengths and potential drawbacks of the proposed model.

**Weaknesses:**

* The major concern is that the benchmark multimodal models are not state-of-the-art. For example, some more recent works also study EHR / clinical note + time series representation learning ([1] Ma, Yingbo, et al. "Global contrastive training for multimodal electronic health records with language supervision." arXiv preprint arXiv:2404.06723 (2024).  [2] Wang, Fuying, et al. "CTPD: Cross-Modal Temporal Pattern Discovery for Enhanced Multimodal Electronic Health Records Analysis." arXiv preprint arXiv:2411.00696 (2024). [3] Cui, Hejie, et al. "Multimodal fusion of ehr in structures and semantics: Integrating clinical records and notes with hypergraph and llm." arXiv preprint arXiv:2403.08818 (2024).). A comparison between the proposed model and more recent multimodal EHR works will strengthen the work significantly.
 * There is no explanation on how the learned embeddings of time series data are used to perform the downstream tasks studied in section 5.2

**Questions:**

* In Figure 3b, the legend on the top right says  "0 < w <= "; something is missing there.
* Equation 1 confuses me. If v and z are not aligned, then how can avgpool of the same kernel size & stride make aligned patches of v and z? For example, if v has a time length of 12 and z of 9, then a kernel of size 3 will give 4 patches of v and 3 patches of z. How are those patches further aligned?
 * In section 5.2, the paragraph says "Decompensation and remaining LoS predictions were made hourly, and we evaluated the remaining LoS predictions in days, as in Sheikhalishahi et al. (2020)." It is very confusing to read, and it will be clearer if the author adds the scale (hourly or daily) of remaining LoS predictions as Decompensation (hourly) in Table 3.
 * The model uses LLM to further embed the time series embedding (Figure 2). Is there any justification for this model design, other than that the AR generation of the next token can be naturally used to recover the time series data?

---

> ### Author Response · Authors · 2025-11-21
> **Response to the reviewer**
>
> We thank the reviewer for acknowledging the novelty of our work and their constructive feedback. We present our point-by-point rebuttal as follows:
>
> ### Regarding the selection of baseline models:
> Given our intended contribution as temporal alignment between two temporal EHR data modalities (Section 2), we didn't focus on comparing LLM4EHR with notes/TS fusion models, as we believe they are less related to our work. In our evaluation, we focused on comparisons with TS models, as we believe we are building on existing work on TS modelling/alignment by introducing a new temporal alignment objective specific to EHR data modalities. However, we agree with the reviewer's suggestion to acknowledge newer works in the text/TS alignment and will adjust Section 2 accordingly. Finally, we note that we couldn't find a peer-reviewed source for one of the suggested references, Ma et al., 2024.
>
> ### Regarding downstream tasks:
> Predictions for downstream tasks are generated by attaching classification/regression heads to these embeddings, as stated in Section 5.1. We'll improve the clarity regarding technical descriptions during editing.
>
> ### In response to specific questions:
>
> **Q1:** We'll fix this part. The original expression was $0 < \omega \leq 1$
>
> **Q2:** We specified that pooling and alignment are done within the temporal dimension; we’ll specify how that changes the padding operation in the appendix. In the reviewer’s specified case, v and z are aligned at the start and end points. So, if v covers a 12 hour period from A to B, z will similarly cover a 12 hour period from A to B. With padding tokens injected to ensure uniform temporal length between v and z. Note that the implication is that padding is done with respect to time, such that if a period of three hours appears in the middle of z, padding will be injected into the missing period and not the end. We explained how this padding process would affect the gradients in our response to reviewer yyS2 Q2. We'll amend the Method sections accordingly.
>
> **Q3:** We’ll rephrase part of the task description during editing.
>
> **Q4:** In our method section, we specified that this was done to ensure the stable optimisation of our alignment objective $\mathcal{L}_{\omega}$. Repurposing frozen LLMs for TS projections have been explored in prior works, which we cited in Section 2.

---

> > ### Comment · Reviewer_2Zsg · 2025-11-21
> >
> > Thanks to the author for the clarification.
> >
> > First, I want to clarify that the reason I cite some of the multimodal EHR literature is that, to my best understanding, clinical notes are also part of EHR, and the work King et al. (2023) is studied in this paper as one of the benchmark EHR models where clinical notes and time series are jointly modeled. Therefore, it gives me the impression that methods working on clinical notes from EHR and time series events from EHR also serve as important contenders. I am also aware that the author identifies "EHR events" as their primary modality to study from the general EHR, rather than the sparse clinical notes. The author may consider additional editing on what the actual most relevant works are to the proposed method.
> >
> > Second, can the author point out which part in section 5.1 states the methodology for performing downstream tasks? Section 5.1 mostly explains empirically how fully labelled datasets are used to simulate actual model deployment. This methodological explanation might be trivial, but it still deserves proper addressing for the completeness of the work.
> >
> > The remaining clarification makes sense, and I agree that the alignment needs better phrasing to deliver the idea stated in the rebuttals better.

---

### Official Review · Reviewer_aL4d · 2025-11-08

**Soundness:** 2
**Presentation:** 2
**Contribution:** 1
**Rating:** 2
**Confidence:** 3

**Summary:**

The paper proposes LLM4EHR, a framework designed to learn joint representations of two modalities in : structured EHR sequences  and numerical clinical time series. The core method is to use the LLM to extract embeddings for  EHR events and to then align them with the time-series data by optimizing a contrastive learning objective. Another innovation is to use the semantic similarity of EHR events to weight the contrastive loss for time series, aiming to mitigate "class collision." The model is evaluated on downstream tasks on MIMIC, eICU, etc and shows superior performance compared to baselines.

**Strengths:**

The core idea of aligning EHR events and time series is a reasonable research direction. Moving beyond instance-wise alignment can be conceptually interesting. Additionally, the problem of better EHR data use is highly relevant. Developing strong foundation models for EHR data has great value for clinical AI. Additionally, the paper is generally well-structured, and the figures pretty illustrative. Finally, I like the few-shot and cross-dataset evaluation of the model.

**Weaknesses:**

Please see questions. Additionally, I am an emergency reviewer, so I have not had the chance to read the paper in detail. If I have misunderstood or missed anything, please bring it to my notice.

**Questions:**

How is the temporal alignment between EHR and the time series achieved/ how are the two modalities reconciled for patch creation.
If a patch contains 6 hours of time series but only 2 EHR events, what is the 'alignment'? What about 1 EHR event?

The model underperforms on the LoS task, which is supposed to be due to 'distortion in embeddings'  Why does a model designed for temporal understanding underperform on temporal regression problem?

Why LLM? If i understand correctly, the LLM is a feature embedder.  Comparing the A.8 results, it seems that the more powerful newer LLMs (like llama) do similar to old models like BERT/RobertA. This does not seem to be a framework which at core rely on a LLM knowledge or ability.
Additionally for models like LLama, how were the embeddings obtained? Is it pooling tokens, using a pretrained MLP, etc. please give details.

Can you add comparisons with a direct baseline that combines the embeddings from a pre-trained EHR LLM (like the one you used) and a pre-trained time series model. Is there other experiments that show this compute-heavy training method is better than such a simpler, more interpretable approach?

---

> ### Author Response · Authors · 2025-11-22
> **Response to the reviewer**
>
> We appreciate the reviewer for their transparency. We acknowledge that some of the reviewers’ concerns stemmed from misunderstandings of the motivation and intended contribution of our paper. We aim to clarify these misunderstandings through our point-by-point rebuttal.
>
> ### On temporal alignment and patch creation:
> We explained in Section 4.1 how alignment is achieved at the patch level. Both EHR and TS embeddings are aggregated into non‑overlapping patches of equal count via average pooling (Eq. 1), as shown in Figure 3. However, we provided more detailed answers regarding specific padding operations and their implications for gradients. We refer the reviewer to our response to reviewer 2Zsg Q2 and yyS2 Q2 for more details. We’ll amend the method sections accordingly.
>
> ### On Remaining Length of Stay (LoS) regression:
> We agree with the reviewer’s observation that LoS results lag specialised TS baselines (In Section 5.2/A.5). The implication of this observation is discussed in detail in both Section 5.2 and Section A.5. We note that LoS regression is one out of the four clinical outcome prediction tasks we included in our evaluation and LLM4EHR improved the performance on all three remaining tasks while maintained consistent performance in a cross-dataset setting. Finally, we note that all four clinical outcome prediction tasks involved in our evaluation requires learning temporal dependencies, as specified by the authors of the clinical benchmarks, we disagree with the reviewer's implication that LoS regression is the only “temporal understanding” task.
>
> ### On the role of LLM
> We appreciate the reviewer’s comment on the motivation of our paper. Relevant prior works that inspired our choice of modelling structure are discussed in detail in the related work section (Section 2, paragraph 2). Additionally, we clarify that the frozen LLM was not used solely as a feature embedder; Figure 2 in Section 4 clearly shows that the frozen autoregressive attention layers of the LLM are used as a consistent mechanism for modelling the transitions of EHR events and TS observations.
>
> The results in A.8, combined with our ablation studies in Section 5.4, show that while larger models provide richer embeddings, our alignment objective contributes most to the performance gains. The observation that domain-specific alignment outweighs model size doesn’t invalidate our hypothesis (Abstract line 016-020).  Additionally, the results in A.8 are consistent with prior research on modelling LLMs via TS (e.g., Liu et al., 2024, cited in Section 2), where domain-specific adaptation outweighs model size.
>
> The embeddings were obtained via pooling tokens of an EHR event’s text descriptor (drug compound, lab component/results). Since similar techniques were used in prior work on modelling EHR events with LLMs (King et al., 2023), we included this information in Section 3.1 rather than the main method section. A more thorough description of technical details will be added to the appendix.
>
> ### On baseline models
> The supervised GPT baseline (Table 2) represents the combination suggested by the reviewer: embeddings from a pre-trained LLM are trained on task supervision directly without our proposed alignment. As shown in Table 3, this baseline consistently underperforms LLM4EHR, demonstrating that joint temporal alignment is more effective than naive combinations of embeddings. A similar baseline (SAnD), with an autoregressive transformer (GPT sans the language weights), was also included. Additionally, our ablation study (Sec. 5.2) shows that removing the alignment objective reduces performance, further supporting the necessity of our approach over simpler alternatives.
>
> We apologise for the belated response. We received a high number of reviews for our paper, and we aim to address individual reviewers' concerns thoroughly. We'll be happy to make additional clarification and engage in discussions with the reviewer.

---

### Official Review · Reviewer_6Vup · 2025-11-10

**Soundness:** 2
**Presentation:** 2
**Contribution:** 2
**Rating:** 2
**Confidence:** 4

**Summary:**

The paper proposes a model for numeric EHR data that integrates text annotations for some observed variables. It proposes a contrastive task to train this model using various learned layers attached to a frozen language model backbone.

**Strengths:**

- Integrating language embeddings into clinical TS models is relevant to study and could boost performance.
- The use of contrastive training between reasonably well-aligned data is promising, given that other multimodal clinical data is challenging to use constrastive training on, due to its poor alignment (e.g., clinical notes versus vitals).

**Weaknesses:**

- The paper presents a separation between clinical time series and EHR entries that's unclear and doesn't reflect the nature of the data. These are not inherently distinct modalities: most EHR entries in the datasets being described are irregular samples of clinical time series, and are used as such in previous work (e.g., EBCL). For another example, the time series features in the Harutyunyan et al. MIMIC-III baseline are derived from chart events and lab events. While it's not clear how this paper understands chart events, it describes lab events as EHR entries even though they contain the same data as in an irregular time series representation. The practical relevance of this model's multimodality is therefore limited.
- I couldn't find basic information about the architecture in the text, including what the "Timeseries embedding" and "Timeseries decoder" blocks in Figure 2 are and how predictions are generated for fine-tuning and inference.
- Hyperparameter tuning for baseline models seems to be missing.
- The main evaluation is limited, being a few-shot prediction with only one fraction of labelled training data evaluated. Full-shot results are not given. Since the tasks being evaluated were generated from EHRs without manual labelling, they aren't the kind of medical modelling tasks where few-shot capabilities are particularly relevant.

**Questions:**

- While using a large language model for text embedding is standard, using one to embed time series with no text information seems awkward, and especially a frozen one. Why not use a time series embedding model instead, or at least train the transformer weights?
- Can you explain the missing elements in Table 3? It's not apparent to me why instances couldn't be constructed for the corresponding models on an hourly basis. While the appendix indicates that EBCL was intended for sequence classification, I would note that their paper does include length-of-stay regression forecasting results.
- Are there cases in the datasets of multiple episodes corresponding to the same patient, and if so, do you ensure that these remain in the same partition?
- How are EHR entries used during fine-tuning and evaluation, for your method and for the other methods?
	- For one, EBCL is designed to use lab events and other features that are discussed as EHR entries in this paper. Were they provided to your EBCL implementation when training and evaluating it?

---

> ### Author Response · Authors · 2025-11-22
> **Response to the reviewer**
>
> We appreciate the reviewer for constructive reviews of our manuscript. We note that some comments appear to stem from misunderstandings of our manuscript and cited materials, and we aim to clarify them via our point-by-point response. Additionally, we would appreciate if the reviewer could elaborate on their specific claims regarding the utility of few-shot evaluations in clinical outcomes predictions.
>
> ### On the separation of modalities:
> In our paper, we define EHR events as discrete and timed actions or occurrences documented within an EHR system. In Section A.2.1 and Section A.2.2, we explained how we extracted and processed lab, procedure and intake events from MIMIC-III and eICU to form EHR sequences. ICD diagnoses were excluded to prevent label leakage in phenotyping.  Conversely, we extracted 11 routinely collected physiological measurements and vital signs as time series (TS) variables, following the clinical benchmarks for MIMIC-III and eICU. We’ve made the distinction clear in multiple sections of our manuscript.
>
> ### On the description of model structure:
> Figure 2 and Section 4 describe the “Timeseries embedding” and “Timeseries decoder” blocks. TS embeddings are produced via frozen autoregressive attention layers, pooled into patches, and decoded autoregressively with a reconstruction loss (Eq. 4). Predictions for downstream tasks are generated by attaching classification/regression heads to these embeddings, as noted in Section 5.1. We acknowledge that other reviewers have asked for clarifications on specific implementation details, which we'll add to our manuscript.
>
> ### On hyperparameter tuning of baseline models:
> Hyperparameter tuning for baseline models were preformed in accordance with the original implementations, we’ll amend the search space for hyperparameters in the appendix section accordingly.
>
> ### On evaluation setting:
> We focus on few‑shot scenarios because labelled ICU data is scarce in practice. Section 5.1 explains that we re‑partitioned datasets to simulate deployment with limited labels. Additionally, we ask the reviewer to clarify what they meant by  “they aren't the kind of medical modelling tasks where few-shot capabilities are particularly relevant”.  We couldn’t find relevant research questioning the utility of few-shot evaluations in general clinical outcome predictions.
>
> ### Regarding specific questions:
>
> **Q1:** We explained the reasoning in detail in Section 2 (relevant prior research) and Section 4 (why frozen LLM attention layers are used).
>
> **Q2:** I would note that ‘Logistic regression’ results in EBCL is different to ‘Linear regression’ results we reported in our paper. Please refer to Section 5.2 (our paper) and Section 3.2 Finetuning (EBCL) for more detail.
>
> **Q3:** Splits are done at patient level, we didn’t specify this because it was explicitly mentioned in the cited benchmark, but we will clarify in the final draft.
>
> **Q4:** We refer the reviewer to our response above. Note that the exact implementation of baseline models are provided in Section A.3.

---

> > ### Comment · Reviewer_6Vup · 2025-11-25
> >
> > Thank you for the response, I've reviewed it and added additional comments below.
> >
> > Regarding the separation of modalities, I am clear on how you define and separate them, my criticism is that this separation is not appropriate to the nature of the data. For instance you might have text events "13:27 Lab event Glucose 127.5; 18:06 Lab event Glucose 127.9; 18:06 Lab event O2Sat 97.0" that trivially map to an irregular multivariate time series, as used in previous papers, and vice versa, at most with an additional text annotation. This is in contrast to other multimodal medical modelling that handles fundamentally different data, like images or free-form text, and is therefore more impactful.
> >
> > To clarify my point about labelled data, a lack of labelled data typically occurs in medical modelling settings where post-hoc manual expert annotations are required. Examples would be labelling X-ray or EKG datasets for different conditions. This does not seem to apply for the benchmarks used, since expert annotations were not required for their labels, they just leveraged attributes that are routinely recorded in EHRs. For instance, it's difficult to imagine a hospital having a set of EHRs for modelling but only having the length of stay available for a small subset, since this is just a function of the recorded discharge date. A full-shot setting is therefore more realistic for this task. It's not clear what realistic tasks we can judge the model's applicability for, given few-shot LoS results pretrained on unlabelled data.
> >
> > Could you please provide an explicit answer to Q4 and the sub-question about EBCL? I'd specifically like to know if the EBCL baseline only used the data extracted as time series variables or whether it used a similarly wide range of EHR events as your model, because this isn't clear to me from Appendix A.3.3.

---

> > > ### Author Response · Authors · 2025-11-26
> > > **Additional response to the reviewer**
> > >
> > > We thank the reviewer for additional clarification on their questions, and here is our point-by-point response to the new discussion raised by the reviewer.
> > >
> > > Regarding the separation of modalities, we can confirm that we did not duplicate events from Lab tables during alignment. We thought that it was obvious from our descriptions, but we will add additional clarifications. Additionally, we presented ablation studies in  Section 5.4 to show the performance gain via our alignment loss, with additional comparison against free-text alignment methods such as King et al in Table 3. We acknowledge that there are inherent benefits to TS alignment with free text or images, and we did not argue otherwise in our paper. We believe we made our motivations and contributions clear in the Introduction and related work.
> > >
> > > Regarding the argument about labelled data, we appreciate the reviewer’s anecdotal observations that few-shot learning is useful for X-ray or EKG datasets, and we did not argue otherwise in our paper. We note that the design of our few-shot evaluation setting is mainly to evaluate our proposed foundation models in task and dataset adaptions (line 289-301), which is common for prior works on clinical foundation models (Section 2 and Wornow et al, 2023), and is consistent with our claim that our work is a step towards building more generalisable and performant clinical foundation models (Abstract). We hope this clears the reviewer’s confusion regarding the proposed applicability of LLM4EHR and why we believe a few-shot evaluation setting would be more appropriate.
> > >
> > > We can understand the reviewer’s question better thanks to the clarification. We used the data definition from EBCL (patient observations as event triplets) and included the same level of EHR events data as used in LLM4EHR. We note that there were ambiguities regarding EBCL’s data usage, as their official implementation did not specify which tables from MIMIC-IV they included (We inferred from Section 4.1.1 of their paper). Here are two specific clarifications that we plan to add to our paper: 1) We used the same level of EHR data in LLM4EHR and EBCL (not just lab observations), except we modelled all EHR observations as observation triplets (time, event, value). 2) We did not include diagnostic codes in both EBCL and LLM4EHR, so as not to leak patient phenotype labels.
> > > Finally, we noted that we’ve made a mistake in Table 2, EBCL was described as an EHR model, but we mislabelled its data modality as TS rather than EHR. This was unintentional and will be fixed.
> > >
> > > We hope that our clarification helped the reviewer to understand our motivations and contributions better. We’ll amend the missing details based on the reviewer’s suggestions. We would appreciate it if the reviewer would consider improving their score if our response is satisfactory.

---

### Official Review · Reviewer_XTGU · 2025-11-25

**Soundness:** 3
**Presentation:** 3
**Contribution:** 3
**Rating:** 4
**Confidence:** 3

**Summary:**

Apologize !
I was assigned as an additional reviewer after the other reviews had already been completed, so I am submitting this review later than the official deadline.
I apologize for this; please consider my comments with the understanding that the authors don't have enough time to fully address them during the rebuttal phase.

The paper proposes LLM4EHR, a multimodal ICU model that uses a shared frozen GPT-2 backbone to encode both structured EHR events and physiologic time series.
Time series are projected into the LLM embedding space, temporally patched, and trained with a combination of InfoNCE, an EHR-guided ω-regularised contrastive loss, and a next-step reconstruction loss.
The model is evaluated via linear probing on multiple ICU prediction tasks and cross-dataset transfer, showing strong gains over TS-only and prior multimodal baselines, especially for classification and domain shift.

**Strengths:**

Conceptually clean architecture: A single frozen LLM shared by EHR and TS with only lightweight projection and head layers makes the approach simple, reusable, and computationally realistic.

Novel EHR-guided contrastive objective: The ω-regularised loss uses EHR semantic similarity to shape TS–TS relations, reducing class collision in InfoNCE and yielding more clinically meaningful TS representations.

Strong robustness and transfer: LLM4EHR consistently outperforms supervised TS models, TS self-supervised methods, and other multimodal baselines on phenotyping, decompensation, and cross-dataset mortality (including adult→pediatric transfer).

Solid ablations: Loss component and backbone ablations clearly support design choices, showing that L_ω mainly drives classification gains while L_recon is important for LOS and preserving numeric TS information.

**Weaknesses:**

1. Unclear whether LLM “knowledge” is truly leveraged.

Although the method is framed as an LLM-based clinical foundation model, in practice the LLM backbone is completely frozen and used purely as a encoder. There is no direct evidence that linguistic/medical knowledge from GPT-2 meaningfully drives the improvements (e.g., no comparison against a similarly sized transformer trained from scratch - SAND may have much smaller embedding dimension size, no analysis of whether EHR token semantics matter beyond providing any transformer encoder).
As a result, it is not entirely convincing that this is really “leveraging an LLM” rather than just using a convenient off-the-shelf backbone.

2. Only linear probing is considered; no end-to-end fine-tuning of GPT-2.

All downstream results are reported under a frozen-encoder, linear-head regime.
Given that GPT-2 at this scale (hidden size 768) is not extremely large by modern standards, it seems feasible to explore at least partial end-to-end fine-tuning (e.g., last few layers, adapters/LoRA), which may have better performance.
Without such experiments, it is hard to judge whether the proposed alignment and loss design remain beneficial once the backbone is allowed to adapt, or whether the gains are specific to the somewhat artificial “representation-only / linear probe” setting.

3. Multimodal and alignment claims remain indirect.

EHR is only used during pretraining and not used at inference (I understand, the number of EHR code used in this work are somewhat limited), and there is no direct embedding-level evidence (e.g., cross-modal retrieval, visualization, similarity analysis) that the model achieves genuine TS–EHR alignment. The observed gains could plausibly be explained by generic TS representation regularization rather than by strong multimodal/LLM effects.

**Questions:**

1. Have you tried partially unfreezing the backbone (last block, adapters, or LoRA) during pretraining, and if so, how does this affect downstream performance, LOS regression, and the stability of the EHR semantic space?

2. In the backbone and baseline comparisons, GPT-2 small uses a hidden size of 768, but the paper does not report the embedding/hidden dimensions or parameter counts for baselines such as SAnD, EBCL, or the TS self-supervised models.
Could you clarify:
(a) the hidden size and total parameter count for each baseline model, and
(b) whether your gains might be partly explained by capacity differences rather than architectural advantages?

3. You may also want to discuss your method in relation to GenHPF (Hur et al., 2022; arXiv:2207.09858), which similarly encodes EHR using a text-based representation, but goes further by converting time-series values into text and using them directly without an explicit alignment stage.
I do not expect a comparison given time and resource constraints, but a short conceptual comparison in the related work or discussion section would help clarify how LLM4EHR differs from this line of text-based EHR/TS modeling.

---

> ### Author Response · Authors · 2025-11-26
> **Response to the reviewer**
>
> We fully appreciate the reviewer for their balanced review of our manuscript. Here is our point-by-point response to the reviewer’s specific questions.
>
> Q1: While we did not attempt a part-by-part unfreezing experiment during our ablation studies, SAnD could be seen as an end-to-end version of our backbone model without the learned GPT 2 language weights. We note that there is a modest performance gain from Supervised GPT to SAnD (both models are comparable in size), and our ablation studies indicate that the major source of performance gain is via our proposed alignment objective.
>
> Q2: a) yes, we can confirm that and will clarify this in the final version of our paper (more details more hyperparameter tuning). b) No, while Section A.6 suggests that there are performance differences between different LLMs as backbones, the majority of our experiments in Section 5 were conducted using GPT 2 as backbone, with improved performance over structurally similar baselines such as SAnD and supervised GPT.
>
> Q3: We thank the reviewer for being reasonable with their suggestions. From the reviewer’s descriptions, we argue that the contribution of GenHPF focused on learning dense embedding of EHR events, while our contribution is on cross-modality alignment between EHR event streams and TS observations. We used a more generic, token-based representation of EHR events demonstrated in prior works (Kraljevic et al.) to isolate the performance impact of our proposed alignment objective. Interestingly, while we argue that our contribution lies in a different field than GenHPF, text-rich dense representations of medical events are an interesting topic and are not fundamentally incompatible with our proposed alignment objective, which could be explored in future works.
>
> Given the large number of reviews we’ve received (this being the 7th), we refer the reviewer to our previous rebuttals for more detailed clarifications on our methods, motivations and planned additions. We hope the reviewer could consider increasing their score if our answers are considered satisfactory.

---

### Author Response · Authors · 2025-11-30
**New results and additional comments (1/n)**

We note that, since we received a large number of reviews for our paper and some reviews arrived late during the discussion period, we didn't have time to incorporate all the new metrics/benchmarks/evaluation scenarios proposed by the reviewers. We apologise for this and the belated results. Additionally, given the recent changes at ICLR, we've decided to post further comments and new results in one single thread instead of replying to individual reviewers.

**Additional few-shot evaluation scenarios**

Reviewer **yyS2** and **qYea** noted that the few-shot claim was weakly justified in our paper. While we argued in Section 5.1 that the few-shot evaluation was more realistic for a real-world deployment setting and made no claims regarding LLM4EHR’s performance in an extremely low-data setting, we find the reviewer’s suggestions for adding additional few-shot scenarios reasonable. We hereby present additional results for the 10% and 5% few-shot scenarios. Additionally, reviewer **2Zsg** asked for additional comparisons against TS/clinical notes alignment models. In addition to our original response to the reviewer, we decided to include CTPD by Wang, Fuying, et al. as an additional baseline model, which we were able to reproduce using their official implementations. We note that we couldn't reproduce the other two models suggested by the reviewer, as they 1) included prompt engineering with Proprietary LLMs, and 2) did not provide the code necessary for reproducing their results. Finally, we added significance testing as suggested by reviewer **qYea** and used **$\ast$** to denote significant results (p<0.05).

Here are the results for fine-tuning using 20% labelled data, with CTPD added, best results are highlighted in bold, $\ast$ denotes statistically significant results (p<0.05):
|      Task      |   Phenotyping   |                 | Decompensation |               | Remaining LOS |                |
|:--------------:|:---------------:|:---------------:|:--------------:|:-------------:|:-------------:|:--------------:|
|      Model     | AUC-ROC (macro) | AUC-ROC (micro) |     AUC-ROC    |    AUC-PRC    |      RMSE     |       R2       |
|       MLP      |  0.622 (0.022)  |  0.728 (0.031)  |  0.816 (0.090) | 0.105 (0.052) | 6.872 (0.028) | -0.237 (0.010) |
|       CNN      |  0.680 (0.005)  |  0.742 (0.005)  |       N/A      |      N/A      |      N/A      |       N/A      |
|      LSTM      |  0.661 (0.005)  |  0.739 (0.004)  |  0.899 (0.033) | 0.309 (0.017) | 5.919 (0.062) |  0.082 (0.019) |
| Supervised GPT |  0.654 (0.004)  |  0.717 (0.021)  |  0.877 (0.009) | 0.235 (0.018) | 6.012 (0.088) |  0.046 (0.095) |
|     SimMTM     |  0.708 (0.004)  |  0.757 (0.033)  |  0.899 (0.015) | 0.325 (0.041) | **5.695 (0.072)** |  **0.156 (0.045)** |
|  PatchTSMixer  |  0.656 (0.003)  |  0.737 (0.003)  |  0.901 (0.008) | 0.328 (0.022) | 5.824 (0.076) |  0.053 (0.025) |
|    PatchTST    |  0.651 (0.002)  |  0.737 (0.003)  |  0.894 (0.015) | 0.280 (0.032) | 5.722 (0.085) |  0.135 (0.027) |
|      EBCL      |  0.694 (0.003)  |  0.743 (0.014)  |       N/A      |      N/A      |      N/A      |       N/A      |
|   King et al   |  0.702 (0.004)  |  0.756 (0.005)  |       N/A      |      N/A      |      N/A      |       N/A      |
|      CTPD     |  0.707 (0.003)  |  0.755 (0.004)  |       N/A      |      N/A      |      N/A      |       N/A      |
|      SAnD      |  0.655 (0.011)  |  0.721 (0.014)  |  0.872 (0.031) | 0.298 (0.045) | 5.873 (0.066) |  0.076 (0.077) |
|     LLM4EHR    |  **0.719 (0.003) $^{\ast}$**  |  **0.761 (0.006) $^{\ast}$**  |  **0.912 (0.003) $^{\ast}$** | **0.333 (0.025)** | 5.743 (0.088) |  0.129 (0.054) |

We clarify that we only reported CTPD's performance in phenotyping since the authors only reported CTPD's performance on mortality and phenotype classifications. Additionally, CTPD was implemented as a sequence classification model with data fusion before classification. Finally, we note that the authors of CTPD did not specify what types of clinical notes were involved, but Figure 1 implied alignment with nursing/progression notes. We did not include additional notes to prevent label leakage during phenotyping, which was not addressed in their paper. We kept the hidden dimensions of CTPD as 768, same as LLM4EHR and performed hyperparameter search on $\lambda_{1}$ (0.1, **0.5**, 1, 2) and $\lambda_{2}$ (0.1, **0.5**, 1, 2) in eq.10 and reported the best phenotyping results. We'll add clarifications on hyperparameter search for all baseline models to our manuscript.

---

> ### Author Response · Authors · 2025-11-30
> **New results and additional comments (2/n)**
>
> Following the previous post, here are the results for fine-tuning using 10% labelled data, with CTPD added, best results are highlighted in bold, $\ast$ denotes statistically significant results (p<0.05):
>
> |      Task      |   Phenotyping   |                 | Decompensation |               | Remaining LOS |                |
> |:--------------:|:---------------:|:---------------:|:--------------:|:-------------:|:-------------:|:--------------:|
> |      Model     | AUC-ROC (macro) | AUC-ROC (micro) |     AUC-ROC    |    AUC-PRC    |      RMSE     |       R2       |
> |       MLP      |  0.550 (0.025)  |  0.620 (0.029)  |  0.784 (0.099) | 0.101 (0.062) | 7.771 (0.055) | -0.260 (0.030) |
> |       CNN      |  0.606 (0.012)  |  0.657 (0.008)  |       N/A      |      N/A      |      N/A      |       N/A      |
> |      LSTM      |  0.590 (0.014)  |  0.659 (0.009)  |  0.865 (0.041) | 0.301 (0.023) | 6.417 (0.101) |  0.077 (0.042) |
> | Supervised GPT |  0.584 (0.009)  |  0.643 (0.019)  |  0.844 (0.015) | 0.229 (0.025) | 6.498 (0.092) |  0.044 (0.095) |
> |     SimMTM     |  0.638 (0.008)  |  0.680 (0.035)  |  0.866 (0.018) | 0.317 (0.046) | 6.137 (0.088) |  **0.149 (0.065)** |
> |  PatchTSMixer  |  0.619 (0.010)  |  0.662 (0.005)  |  0.868 (0.012) | 0.320 (0.026) | 6.275 (0.084) |  0.051 (0.055) |
> |    PatchTST    |  0.628 (0.008)  |  0.663 (0.005)  |  0.863 (0.021) | 0.273 (0.037) | 6.140 (0.092) |  0.129 (0.048) |
> |      EBCL      |  0.630 (0.007)  |  0.668 (0.012)  |       N/A      |      N/A      |      N/A      |       N/A      |
> |   King et al   |  0.646 (0.010)  |  0.691 (0.006)  |       N/A      |      N/A      |      N/A      |       N/A      |
> |      CTPD     |  0.630 (0.009)  |  0.694 (0.005)  |       N/A      |      N/A      |      N/A      |       N/A      |
> |      SAnD      |  0.600 (0.014)  |  0.664 (0.017)  |  0.846 (0.028) | 0.292 (0.037) | 6.147 (0.075) |  0.073 (0.079) |
> |     LLM4EHR    |  **0.664 (0.009) $^{\ast}$**  |  **0.702 (0.012) $^{\ast}$**  |  **0.885 (0.006) $^{\ast}$** | **0.326 (0.026)** | **6.006 (0.099)** |  0.125 (0.067) |
>
> Here are the results for fine-tuning using 5% labelled data, with CTPD added, best results are highlighted in bold, $\ast$ denotes statistically significant results (p<0.05):
>
> |      Task      |   Phenotyping   |                 | Decompensation |               |  Remaining LOS |                |
> |:--------------:|:---------------:|:---------------:|:--------------:|:-------------:|:--------------:|:--------------:|
> |      Model     | AUC-ROC (macro) | AUC-ROC (micro) |     AUC-ROC    |    AUC-PRC    |      RMSE      |       R2       |
> |       MLP      |  0.380 (0.022)  |  0.424 (0.032)  |  0.592 (0.090) | 0.080 (0.058) | 10.885 (0.070) | -0.479 (0.028) |
> |       CNN      |  0.456 (0.019)  |  0.467 (0.010)  |       N/A      |      N/A      |       N/A      |       N/A      |
> |      LSTM      |  0.463 (0.016)  |  0.484 (0.011)  |  0.716 (0.045) | 0.244 (0.025) |  9.204 (0.110) | -0.142 (0.039) |
> | Supervised GPT |  0.461 (0.010)  |  0.472 (0.019)  |  0.716 (0.020) | 0.186 (0.027) |  8.899 (0.078) | -0.154 (0.195) |
> |     SimMTM     |  0.524 (0.012)  |  0.503 (0.038)  |  0.745 (0.025) | 0.267 (0.047) |  **8.293 (0.092)** | -0.033 (0.077) |
> |  PatchTSMixer  |  0.511 (0.010)  |  0.490 (0.007)  |  0.749 (0.022) | 0.274 (0.029) |  9.080 (0.080) | -0.108 (0.063) |
> |    PatchTST    |  0.533 (0.009)  |  0.491 (0.008)  |  0.745 (0.022) | 0.235 (0.040) |  8.796 (0.090) | -0.054 (0.054) |
> |      EBCL      |  0.535 (0.009)  |  0.508 (0.015)  |       N/A      |      N/A      |       N/A      |       N/A      |
> |   King et al   |  0.554 (0.013)  |  0.552 (0.012)  |       N/A      |      N/A      |       N/A      |       N/A      |
> |      CTPD     |  0.542 (0.012)  |  0.570 (0.011)  |       N/A      |      N/A      |       N/A      |       N/A      |
> |      SAnD      |  0.521 (0.015)  |  0.551 (0.021)  |  0.740 (0.030) | 0.259 (0.040) |  8.697 (0.078) | -0.073 (0.082) |
> |     LLM4EHR    |  **0.577 (0.010) $^{\ast}$**  |  **0.588 (0.014) $^{\ast}$**  |  **0.778 (0.012) $^{\ast}$** | **0.293 (0.028)** |  8.293 (0.090) | **-0.027 (0.070)** |
>
> We note that the additional baseline and few-shot evaluations further support our few-shot claim, LLM4EHR consistently outperformed baseline models at all data partitions with significant gains in classification tasks (phenotyping and decompensation). Notably, the margin of improvement for phenotyping remained significant across all three data partitions, further demonstrating LLM4EHR's robustness. Although LLM4EHR outperformed baseline models in LoS regression at 10% data partition, the gain is not significant, which agrees with our prior conclusion that LLM4EHR is better at classifications than regression. Finally, we note that the change in decompensation AUC-PRC is not significant across all data partitions as decompensation is dominated by negative labels due to being an hourly mortality forecast.

---

> > ### Author Response · Authors · 2025-11-30
> > **New results and additional comments (3/n)**
> >
> > Finally, we reiterate a part of our earlier response to reviewer **XTGU**. The reviewer suggested additional fine-tuning experiments for LLM4EHR, but we misinterpreted the comments as requesting additional baseline models. We clarify that we didn't include additional tuning experiments beyond linear probing, as our contributions are not related to novel tuning methods. However, the reviewer's concern that LLM4EHR is specific to linear probing is valid, and we clarify here that LLM4EHR is natively adaptable to several fine-tuning methods such as end-to-end, bottleneck adapters and prefix tuning. Here, we present additional experiments for fine-tuning LLM4EHR:
> >
> > |        Task        |   Phenotyping   |                 | Decompensation |               | Remaining LOS |               |
> > |:------------------:|:---------------:|:---------------:|:--------------:|:-------------:|:-------------:|:-------------:|
> > |       Tuning       | AUC-ROC (macro) | AUC-ROC (micro) |     AUC-ROC    |    AUC-PRC    |      RMSE     |       R2      |
> > |       Linear       |  0.719 (0.003)  |  0.761 (0.006)  |  0.912 (0.003) | 0.333 (0.025) | 5.743 (0.088) | 0.129 (0.054) |
> > |     End-to-end     |  **0.748 (0.005)**  |  **0.791 (0.006)**  |  **0.939 (0.004)** | **0.343 (0.015)** | 5.687 (0.072) | 0.130 (0.049) |
> > |    adapter (64)    |  0.730 (0.002)  |  0.773 (0.003)  |  0.920 (0.004) | 0.336 (0.018) | 5.643 (0.074) | 0.154 (0.045) |
> > |    adapter (96)    |  0.734 (0.003)  |  0.777 (0.003)  |  0.930 (0.003) | 0.339 (0.019) | 5.583 (0.069) | 0.161 (0.043) |
> > |    adapter (128)   |  0.734 (0.002)  |  0.777 (0.004)  |  0.934 (0.003) | 0.341 (0.021) | **5.432 (0.071)** | **0.172 (0.045)** |
> > | prefix (20 tokens) |  0.738 (0.004)  |  0.781 (0.004)  |  0.919 (0.004) | 0.335 (0.022) | 5.750 (0.080) | 0.128 (0.049) |
> >
> > The above results are produced under 20% tuning data partition, same as our other ablation experiments in Section 5. The numbers in brackets for tuning methods indicate the size of the adapter bottleneck and the number of prefix tokens (suffix in our implementation, as we added tunable vectors after the main TS sequences). As suggested by the new results, LLM4EHR is not limited to linear probing and supports additional tuning methods; some are more suitable for specific tasks, such as bottleneck adapters for LoS regression. We'll add this part as an additional part to our main manuscript and amend the discussion accordingly. Finally, we note that we kept linear probing as the tuning method for comparison with other baseline models to remain consistent with our original claims.

---

### Meta-Review · Area_Chair_5gyQ · 2025-12-28

**Summary:**

The paper proposes LLM4EHR, a framework to align clinical time series with medical event sequences using a frozen LLM backbone. The method introduces a regularized contrastive objective to handle "class collision" by utilizing the semantic similarity of EHR events. The model is evaluated on MIMIC-III, eICU, Physio2012, and PICU for tasks including mortality prediction, phenotyping, decompensation, and length of stay (LoS) regression. The authors position this as a "clinical foundation model" capable of few-shot learning and cross-dataset transfer.

While the idea of aligning EHR modalities is relevant, the method appears to be an incremental combination of existing techniques without sufficient evidence that the LLM's capabilities are truly driving the performance. The performance trade-offs (poor LoS regression) and the artificial separation of time series and EHR events further limit the paper's contribution.

**Reviewer Concerns:**

Addressed Concerns:

- The authors provided additional results for few-shot scenarios (1%, 5%, 10%) and included the CTPD baseline as requested. They also provided results for adapter and prefix tuning to address concerns that the method was limited to linear probing.

- The authors clarified the implementation of the patching mechanism and the handling of empty EHR patches.





Outstanding Concerns:

- A critical concern raised by Reviewer 6Vup is that the distinction between "clinical time series" and "EHR events" in this specific context is artificial and does not reflect the inherent nature of the data, as many EHR events are sparse time series. This undermines the fundamental premise of the "multimodal" alignment contribution. The experimental results of ICU data only on those limited tasks are not very generalizable for real clinical settings.

- Multiple reviewers (XTGU, aL4d) questioned the necessity and utility of the frozen LLM backbone. There is a lack of convincing evidence that the "linguistic knowledge" of the LLM is being leveraged, or if the performance gains come simply from the contrastive objective itself. The method uses the LLM essentially as a feature embedder, and newer/larger LLMs did not yield significant improvements over older ones.

- Multiple Reviewers viewed the work as an incremental combination of established contrastive methods and LLM embeddings rather than a new paradigm.

- The model consistently underperforms on the Length of Stay (LoS) regression task compared to specialized baselines. While the authors argue this is a trade-off for classification alignment, it weakens the claim of a general-purpose "clinical foundation model".

**Reviewer Scores:**

I don't think reviewers will change their scores.

---

### Decision · Program_Chairs · 2026-01-26

Reject